# Divergent single cell transcriptome and epigenome alterations in ALS and FTD patients with C9orf72 mutation

Junhao Li [1,6], Manoj K. Jaiswal [2,6], Jo-Fan Chien [3], Alexey Kozlenkov[2], Jinyoung Jung[2], Ping Zhou[2], Mahammad Gardashli[4], Luc J. Pregent[4], Erica Engelberg-Cook [4], Dennis W. Dickson [4], Veronique V. Belzil [4], Eran A. Mukamel [1] ✉ & Stella Dracheva [2,5] ✉

A repeat expansion in the C9orf72 (C9) gene is the most common genetic cause of amyotrophic lateral sclerosis (ALS) and frontotemporal dementia (FTD). Here we investigate single nucleus transcriptomics (snRNA-seq) and epigenomics (snATAC-seq) in postmortem motor and frontal cortices from C9-ALS, C9-FTD, and control donors. C9-ALS donors present pervasive alterations of gene expression with concordant changes in chromatin accessibility and histone modifications. The greatest alterations occur in upper and deep layer excitatory neurons, as well as in astrocytes. In neurons, the changes imply an increase in proteostasis, metabolism, and protein expression pathways, alongside a decrease in neuronal function. In astrocytes, the alterations suggest activation and structural remodeling. Conversely, C9-FTD donors have fewer high-quality neuronal nuclei in the frontal cortex and numerous gene expression changes in glial cells. These findings highlight a context-dependent molecular disruption in C9-ALS and C9-FTD, indicating unique effects across cell types, brain regions, and diseases.

Amyotrophic lateral sclerosis (ALS) is a fatal neurodegenerative disease that manifests in progressive loss of cortical, brainstem and spinal motoneurons resulting in paralysis, and eventually death, typically, within 3–5 years of symptoms onset[1]. In ~10% of ALS patients, there is a clear family history of the disease, and pathogenic mutations are found in more than half of these cases[2]. By contrast, apparently sporadic ALS is considered a complex trait with an estimated heritability of 40–50%[3]. The pathology and genetics of ALS overlap with another neurodegenerative disease, frontotemporal dementia (FTD), which is the second most common cause of dementia in patients under 65 years[4]. Aggregates of the RNA-binding transactive response DNA binding 43 protein (TDP-43) accumulate in neurons of almost all ALS cases and

approximately half of FTD cases[5], and ALS and FTD can occur within the same family and even in the same person[6].

Mutations in several genes are causative of ALS, FTD, or ALS/FTD[7]. The most common genetic cause is a dynamic hexanucleotide (GGGGCC; $G_4C_2$) repeat expansion in the first intron of the C9orf72 (C9) gene, which is found in ~40% of familial and ~12% of all ALS and FTD cases[8–10]. The non-coding $G_4C_2$ expansion appears to have two direct consequences: a loss-of-function that causes C9orf72 haploinsufficiency, and a gain-of-function associated with the expression of abnormal, bidirectionally transcribed RNAs containing the repeat[11]. These RNAs accumulate as RNA foci and/or are translated into toxic dipeptide repeat proteins (DPRs)[12–14]. ALS- and FTD-associated genes (including C9orf72) are expressed in multiple neuronal and

[1]Department of Cognitive Science, University of California San Diego, La Jolla, CA 92037, US. [2]Friedman Brain Institute and Department of Psychiatry, Icahn School of Medicine at Mount Sinai, New York, NY 10029, US. [3]Department of Physics, University of California San Diego, La Jolla, CA 92037, US. [4]Department of Neuroscience, Mayo Clinic, Jacksonville, FL 32224, US. [5]Research & Development and VISN2 MIREC, James J, Peters VA Medical Center, Bronx, NY 10468, US. [6]These authors contributed equally: Junhao Li, Manoj K. Jaiswal. ✉e-mail: emukamel@ucsd.edu; stella.dracheva@mssm.edu

non-neuronal cell types in the brain and spinal cord[15], raising the question of what impact the mutations have in each individual cell type.

ALS is caused by the degeneration of upper and lower motor neurons, affecting the giant pyramidal cells (Betz cells) in layer 5 of primary motor cortex and the large multipolar alpha motor neurons of the brainstem and spinal cord, whereas FTD results mainly from the degeneration of the large bipolar Von Economo neurons located in layer 5 of several specific cortical regions[16–18]. Animal and in vitro models associated multiple motoneuron-intrinsic (cell-autonomous) pathways with ALS pathogenesis, including glutamate excitotoxicity, mitochondrial dysfunction, RNA metabolism, nucleocytoplasmic transport, and alterations in protein homeostasis[19]. Although these pathways responded to therapeutic modulation in experimental models, clinical translation has been slow. The only approved therapies for ALS (riluzole, edaravone, and relyvrio) prolong survival by only several months[20], suggesting limitations of current disease models, especially those uniquely focusing on motoneurons. Other neuron types[21], as well as astrocytes and microglia, have been implicated in the pathogenesis of ALS and FTD[22,23].

We used single-nucleus transcriptomic (snRNA-seq) and epigenomic (snATAC-seq) analysis of human brain samples to investigate the cell-type-specific molecular alterations in C9-ALS or C9-FTD donors. In both diseases, we studied two key cortical regions, the motor and frontal cortices, in an attempt to uncover regional differences. In C9-ALS, we found the most pronounced transcriptional disruption in upper layer excitatory neurons and astrocytes, and parallel disruption of the epigenome, with concordant changes in chromatin accessibility, histone modifications, and gene expression in specific cell types. Our comparative analysis of C9-ALS and C9-FTD highlights distinct molecular pathologies in these two neurodegenerative diseases.

## Results

### Single nucleus transcriptomes in C9-ALS and C9-FTD specimens identify fine-grained cortical cell types

We performed snRNA-seq on nuclei isolated from the motor cortex (Brodmann area 4) and dorsolateral prefrontal cortex (Brodmann area 9; hereafter "frontal cortex") of autopsied C9-ALS ($n = 6$), C9-FTD ($n = 5$), and pathologically normal (hereafter "controls", $n = 6$) brains (Supplementary Dataset 1). We obtained 105,120 high-quality nuclear gene expression profiles (45,376 from C9-ALS, 9445 from C9-FTD, and 50,299 from controls), detecting a median of 6351 unique mRNA molecules and 2665 genes per nucleus, indicating the high quality of the data. Far fewer nuclei passed quality control in C9-FTD (median 24.3%) compared with controls (66.2%) or C9-ALS (68.4%) (two-sided Welch's $t$-test, $p < 0.05$; Supplementary Fig. 1a–c; see also "A distinct signature of glial dysregulation in C9-FTD" below).

We used iterative clustering to identify 49 distinct subpopulations of excitatory neurons, inhibitory neurons and non-neuronal cells (Fig. 1a, b; Supplementary Dataset 2). The clustering was not biased by brain region, donor sex, sequencing batch, individual samples, or diagnosis (Fig. 1c, Supplementary Fig. 2a–e), and the resulting clusters were stable with respect to the resolution parameter for community detection (Supplementary Fig. 2f). The subpopulations, which we annotated based on the expression of known marker genes (Supplementary Fig. 3), were highly consistent with a recent large-scale study of the human motor cortex (Supplementary Fig. 4)[24]. These results demonstrated the fine-grained cell subtype resolution of our dataset. However, to leverage the information shared across multiple donors when estimating the molecular dysregulation in C9-ALS and C9-FTD, a statistically meaningful number of cells from each donor is required. Therefore, we focused our analysis on 14 major cell types (8 neuronal, 6 glial), defined by grouping subpopulations sharing well-defined markers (Supplementary Fig. 3). At this level of cell type resolution, we observed consistent expression patterns across donors in each major cell type and brain region of the control donors (Fig. 1d).

### Disruption of gene expression in C9-ALS astrocytes and excitatory neurons

The rare giant pyramidal Betz cells in layer 5b of the human primary motor cortex are considered especially vulnerable in ALS[25,26]. Yet, other excitatory and inhibitory neurons, as well as other cortical areas such as the frontal regions, have also been reported to be affected in the disease[21,27]. Global analyses using principal components, Augur[28], and linear discriminant analysis demonstrate significant differences between disease and control samples (Supplementary Figs. 14, 15). To identify genes dysregulated in C9-ALS in a cell type- and region-specific manner, we compared nuclear transcriptional profiles from C9-ALS and control subjects using a mixed effects model[29], while adjusting for demographic and technical covariates (Methods).

Excitatory neurons in all cortical layers were profoundly affected in C9-ALS, with hundreds of differentially expressed (DE) genes in each cell type (FDR < 0.05) (Supplementary Dataset 3, 4). There were more DE genes in upper layer excitatory neurons (L2/3) compared with excitatory neurons in other layers (Fig. 2a). In particular, there were 2.48-fold more DE genes in upper- compared with deep-layer (L5/6) excitatory neurons in motor cortex; in frontal cortex, the enrichment was 2.69-fold. This difference was evident even after subsampling each cell type to ensure equal statistical power for detecting DE genes (Fig. 2b). We also found that deep layer excitatory neurons were more affected in motor than in frontal cortex, with a larger number of DE genes after controlling for sample size (Fig. 2b). Inhibitory neurons had significantly fewer DE genes. Among non-neuronal cells, astrocytes were the most strongly affected in both brain regions (Fig. 2a, b).

Overall, transcriptional changes in C9-ALS were highly consistent in motor and frontal cortices (Pearson $r = 0.49–0.74$, Fig. 2c). Cluster analysis of the most strongly affected DE genes (fold-change > 2; Fig. 2d) showed a unique set of upregulated (cluster C3) or downregulated (C6) genes in astrocytes. In contrast, transcriptional changes in neurons were either shared between multiple neuronal subtypes (Clusters C2 and C5) or specific for excitatory neurons (Cluster C4).

We validated our snRNA-seq findings using bulk RNA-seq measurements in purified nuclei from major cortical cell types (Supplementary Dataset 1, Methods). Using fluorescence-activated nuclear sorting (FANS), we purified neurons, oligodendrocyte lineage cells (mature oligodendrocytes and oligodendrocyte precursor cells, OPCs), and non-oligodendrocyte glial cells. The third group mainly consisted of astrocytes and microglia (Supplementary Fig. 5a, b; Methods). Differential gene expression fold-changes in the single nucleus were strongly correlated with FANS RNA-seq data for corresponding cell types in both cortical regions (Fig. 2e), especially for astrocytes in motor cortex (Spearman $r = 0.517$, Fig. 2f).

Studies in bulk brain tissues showed that the hexanucleotide repeat expansion in *C9orf72* lowers expression of this gene[9,30]. Using our single cell data, we found the highest expression of the *C9orf72* gene in neurons, with comparable expression levels in all neuronal subtypes (Fig. 2g). *C9orf72* was downregulated in C9-ALS excitatory, but not inhibitory neurons (FDR < 0.05). Among glial cells, only astrocytes and oligodendrocytes showed lower *C9orf72* expression in C9-ALS. However, because C9-ALS and control samples differ by both genotype and disease status, our study cannot determine whether downregulation of *C9orf72* expression in specific cell types is directly caused by the repeat expansion mutation, or is also influenced by ALS disease processes.

To connect transcriptional alterations in C9-ALS to protein expression, we used automated Western blotting in bulk motor cortex tissue (Methods). We tested multiple antibodies for proteins that were encoded by DE genes. Antibodies against 19 DE gene products consistently detected a protein band of the expected size in control samples. Using these antibodies, we confirmed the predicted changes in protein levels for 6 of the 19 DE genes, with no discordant changes observed for any of the tested proteins (Figs. 3, 4; Supplementary Figs. 6, 7; Supplementary Dataset 10).

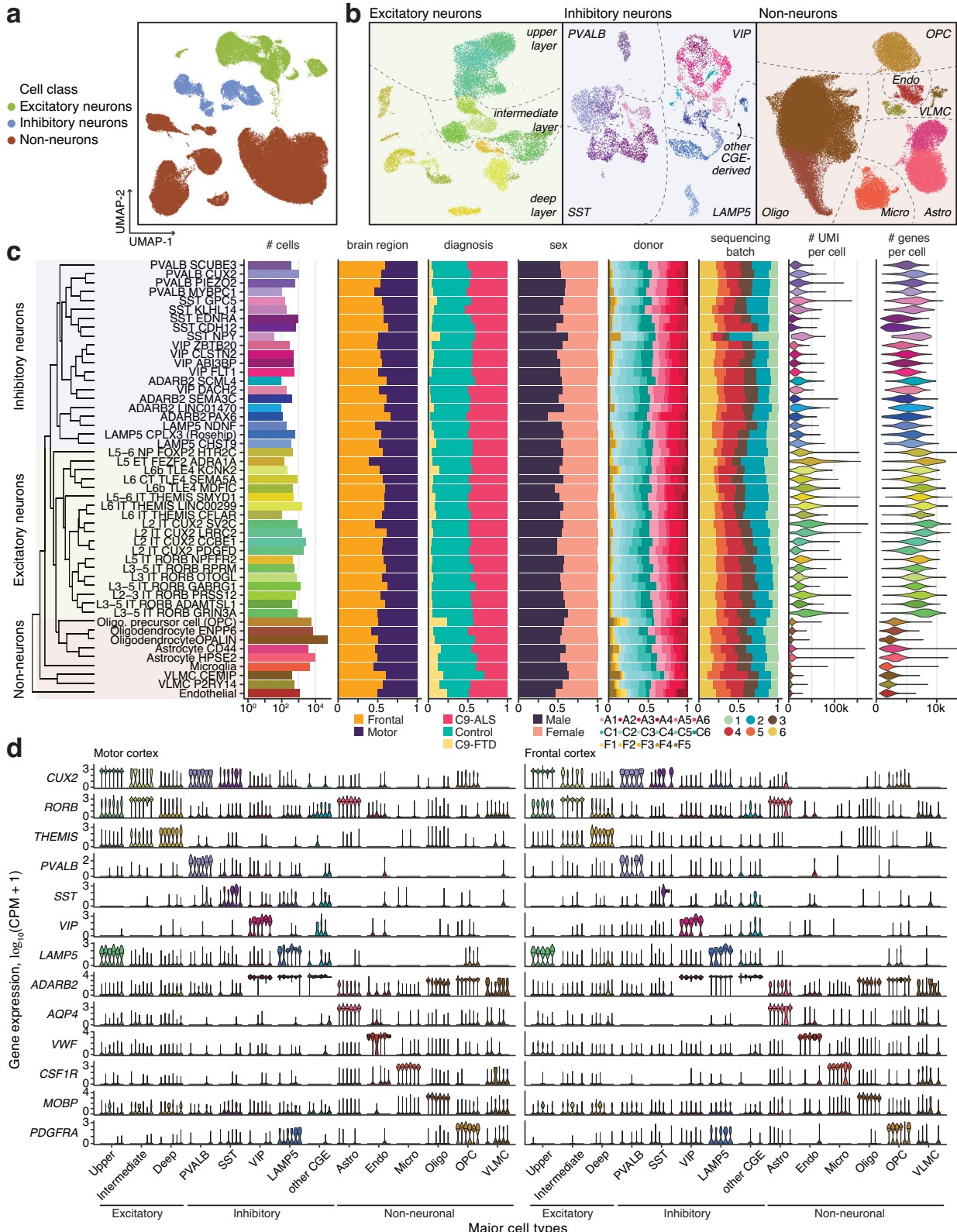

**Fig. 1 | A single cell transcriptomic analysis of human C9-ALS and C9-FTD motor and frontal cortex. a, b** Identification of transcriptomic cell types from snRNA-seq (*n* = 105,120 cells from 17 donors). Nuclei were first classified into three cell classes in (**a**), then subpopulations in each class were identified in (**b**). CGE caudal ganglionic eminence, Astro astrocytes, Endo endothelial cells, Micro microglia, Oligo oligodendrocytes, OPC oligodendrocyte precursor cells, VLMC vascular leptomeningeal cells, IT intratelencephalic, CT corticothalamic, NP near-projecting, ET extratelencephalic. See Supplementary Fig. 3 for marker genes used in the cell type annotation. **c** Neuronal and non-neuronal cell types were distributed across brain regions, donor diagnosis groups, sex, and sequencing batches. UMI unique molecular identifier. **d** Violin plots show marker gene expression in major cell types in each of the 6 control donors. CPM counts per million. Source data are provided as a Source Data file.

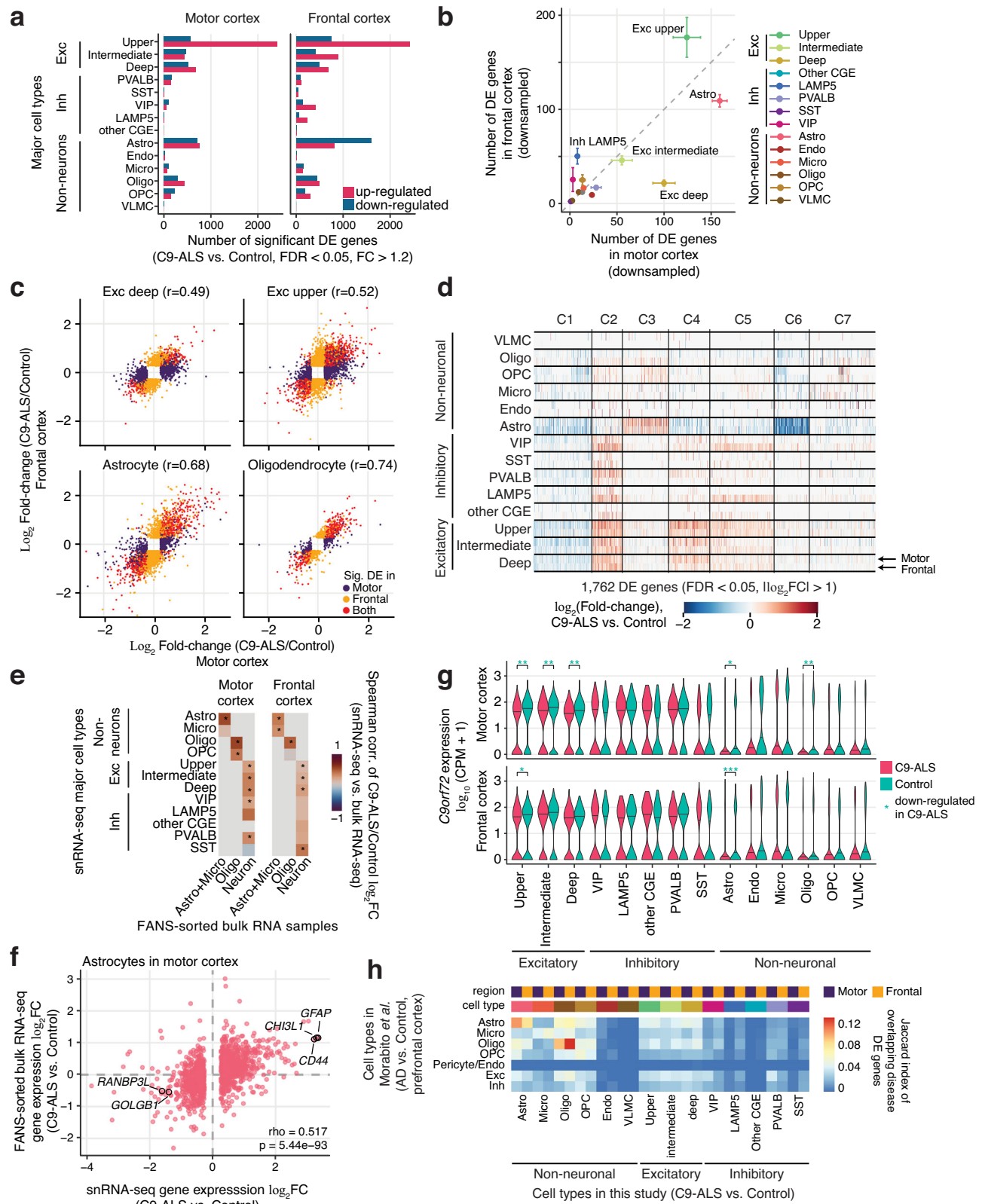

To examine whether our findings were specific to C9-ALS or were general responses to a degenerative process, we compared the DE genes in our C9-ALS vs. control analysis with the reported DE genes in Alzheimer's disease (AD) in prefrontal cortex[31]. We found little overlap between the C9-ALS DE genes and the AD DE genes (the Jaccard index <0.14 for all comparisons) (Fig. 2h). The overlap was highest for glial cell types (astrocytes and oligodendrocytes), suggesting a shared

component of transcriptional disruption in glia across these two neurodegenerative diseases. The differences in neurons in C9-ALS or AD were more distinct. For genes that were significantly DE in both diseases, the fold-changes were in general correlated between the two diseases ($r > 0.58$, $p < 1E-14$ for astrocytes and oligodendrocytes, and $r > 0.23$, $p < 0.006$ for neuronal subtypes in prefrontal cortex) (Supplementary Fig. 13a). We also examined the significant DE genes

**Fig. 2 | Dysregulation of gene expression in C9-ALS is concentrated in astrocytes and excitatory neurons. a** The number of differentially expressed (DE) genes (fold-change>1.2, FDR < 0.05). Exc, excitatory neurons; Inh, inhibitory neurons. **b** After downsampling to 30 nuclei per donor in each cell type to ensure equivalent statistical power, the majority of DE genes were detected in astrocytes and excitatory neurons. Astrocytes and deep layer excitatory neurons were more affected in motor cortex compared with frontal cortex. Random downsampling was performed 10 times, and the dots and error bars represent mean ± SEM. **c** C9-ALS transcriptional differences were consistent in two cortical regions. r, Pearson correlation coefficient. **d** K-mean clustering analysis of strongly DE genes (>2 fold-change) showed distinct groups of genes are affected in neurons and astrocytes. **e** Validation of snRNA-seq differential expression by bulk RNA-seq in FANS-purified

cells. Spearman correlations of the C9-ALS vs. control FC between the snRNA-seq and bulk-RNA-seq were computed using significant DE genes found in snRNA-seq shown in (**a**). *Spearman correlation FDR < 0.05. **f** C9-ALS astrocytes exhibit consistent differential expression in snRNA-seq and in bulk RNA-seq. Dots represent significant DE genes (FC > 1.2, FDR < 0.05) found in snRNA-seq. Selected genes of interest with concordant FC are highlighted. rho, Spearman correlation coefficient. **g** *C9orf72* expression was downregulated in C9-ALS in excitatory neurons, astrocytes and oligodendrocytes. Asterisks indicate significant differential expression identified with MAST (*FDR < 0.05; **FDR < 0.01; ***FDR < 0.001). CPM, counts per million. **h** Overlaps of significant C9-ALS vs. control DE genes (this study) and AD vs. control DE genes (Morabito et al.) for each cell type, measured in Jaccard index. Source data are provided as a Source Data file.

that had concordant or discordant effects in ALS (our study) vs. AD[31]. For this comparison, we focused on the cell types that were most affected in C9-ALS, such as astrocytes and excitatory upper/deep neurons, as well as oligodendrocytes that had the highest Jaccard index (Supplementary Fig. 13b−e). We found that *CD44* and *MAOB* were upregulated in astrocytes in both C9-ALS and AD, whereas *DNMT3A* was downregulated and *HSPH1* was upregulated in both neuronal subtypes. In contrast, *KNDC1* was upregulated in C9-ALS but downregulated in AD in prefrontal cortex astrocytes, whereas *HTR1E* was downregulated in C9-ALS but upregulated in AD in the prefrontal cortex deep excitatory neurons.

## Astrocyte transcriptional dysregulation in C9-ALS

We observed a distinct transcriptional signature of astrocyte activation in C9-ALS brains[32]. mRNA for *GFAP*, a classic marker of astrocyte activation that encodes an intermediate filament protein, increased 10.3-fold in astrocytes of the motor cortex, and 2.4-fold in frontal cortex (Fig. 3a, b). Likewise, we found higher expression of the reactive astrocyte-associated genes *CD44* (9.4-fold in motor cortex, 3.7-fold in frontal) and *CHI3L1* (10-fold in both regions). CD44-positive astrocytes were found in a variety of neurological diseases[33], and elevated levels of chitinases (including CHI3L1) have been proposed as a biomarker of ALS severity and progression[34,35]. We confirmed that GFAP and CD44 proteins were upregulated in C9-ALS bulk motor cortex tissue (*p* < 0.05; Fig. 3c; Supplementary Fig. 6). Furthermore, immunofluorescence confirmed that GFAP protein is highly enriched throughout the cell body and processes of astrocytes in C9-ALS motor cortex (Fig. 3d).

To assess the functional relevance of the altered gene expression in astrocytes, we performed functional enrichment analysis of DE genes[36]. In both cortices, upregulated genes were enriched for multiple functions including cell adhesion and extracellular matrix (exemplified by *ITGA6-7, CLU, COL4A2* genes), actin cytoskeleton (e.g., *ACTN1, PALLD*), and cell migration (e.g., *CRB2, CORO1C*), suggesting substantial cytoskeleton and cell-surface protein remodeling of the C9-ALS astrocytes (GO enrichment FDR < 0.05, Fig. 3e; Supplementary Dataset 5). DE genes in these functional categories were affected in a consistent manner in both motor and frontal cortices (*r* > 0.7, Fig. 3f, Methods). Remodeling of astrocyte morphology is a hallmark of neurodegeneration[37], and our findings highlight the genes involved in this process in C9-ALS. We did not find any functional categories enriched for genes that were downregulated in astrocytes.

We detected changes in the expression of genes whose protein products or functional pathways were previously associated with ALS. For example, several studies reported alterations of the transforming growth factor-β2 (TGF-β) signaling in ALS models[38,39]. Consistent with these reports, we found strong upregulation of the *TGFB2* gene (1.56-1.95 fold in the two cortices), as well as an increased TGF-β protein level in C9-ALS astrocytes (Fig. 3a−c, Supplementary Fig. 6). We found that the expression of *RANBP3L*, a gene encoding a Ran-binding protein involved in nucleocytoplasmic transport[40], was downregulated 3-fold in motor cortex of C9-ALS astrocytes (Fig. 3a, b). We also detected a

trend to a reduction of RANBP3L protein in the C9-ALS motor cortex (*p* = 0.063) (Fig. 3c, Supplementary Fig. 6). This finding is consistent with reports of reduced expression of neuronal nucleocytoplasmic transport proteins in C9-ALS[41–44].

Our data enabled a granular cell-type-specific analysis of transcriptional alterations in C9-ALS, in some cases, identifying genes with opposite patterns of altered expression in different cell types. For example, we found that *ATP2B4*, which encodes plasma membrane calcium ATPase PMCA4 that catalyzes the hydrolysis of ATP coupled with transport of Ca$^{2+}$ from the cytoplasm to the extracellular milieu, showed ~2.0-fold higher expression in C9-ALS *vs.* control astrocytes in both cortical regions (Fig. 3a, b). In contrast, *ATP2B4* was downregulated in upper and deep layer excitatory neurons. Astrocytes exhibit dynamic cell state changes mediated by cytosolic Ca$^{2+}$[45]. Thus, increased expression of *ATP2B4* in astrocytes likely represents a protective mechanism to counterbalance upregulation of Ca$^{2+}$ levels in C9-ALS.

## C9-ALS-associated transcriptional alterations in excitatory neurons

Our analysis showed that upper layer (L2/3) and deep layer (L5/6) excitatory neurons were strongly affected in C9-ALS (Fig. 2b), and we found a large number of DE genes in both cortical regions (examples in Fig. 4a; Supplementary Dataset 3, 4). Functional analysis of genes upregulated in motor cortex showed the most significant enrichments for gene ontology (GO) categories associated with mitochondrial function, protein synthesis, and cellular proteostasis, the process by which the health of the cell's proteins is monitored and maintained[46] (FDR < 0.05) (Fig. 4c; Supplementary Dataset 5). There were also notable enrichments for categories associated with nucleocytoplasmic transport and DNA damage.

Although C9-ALS upper and deep excitatory neurons showed a largely consistent effect on gene expression in motor and frontal cortices (Fig. 2c), only a small number of GO categories were enriched for genes upregulated in frontal cortex (Supplementary Dataset 5). To understand the regional differences of the disrupted pathways, we compared genes from the enriched GO categories across the two regions (Fig. 4c−e; Methods). We found a similar pattern of differential expression for genes associated with cellular proteostasis (protein folding) in both regions. However, genes associated with mitochondrial function (respiratory chain) were more affected in motor compared to frontal cortex in deep layer excitatory neurons. Genes associated with protein synthesis (protein localization to endoplasmic reticulum) were more affected in motor cortex in both neuronal subtypes. This observation was consistent with our downsampling analysis (Fig. 2b), showing that certain genes in deep layer excitatory neurons were more strongly affected in C9-ALS in motor cortex compared to frontal cortex. The majority of the DE genes showed stronger upregulation in upper *vs.* deep layer excitatory neurons (Fig. 4f).

We detected widespread upregulation of mitochondrial genes in C9-ALS (Fig. 4c, d). Among the DE genes, were those coding for subunits of all four mitochondrial respiratory chain complexes

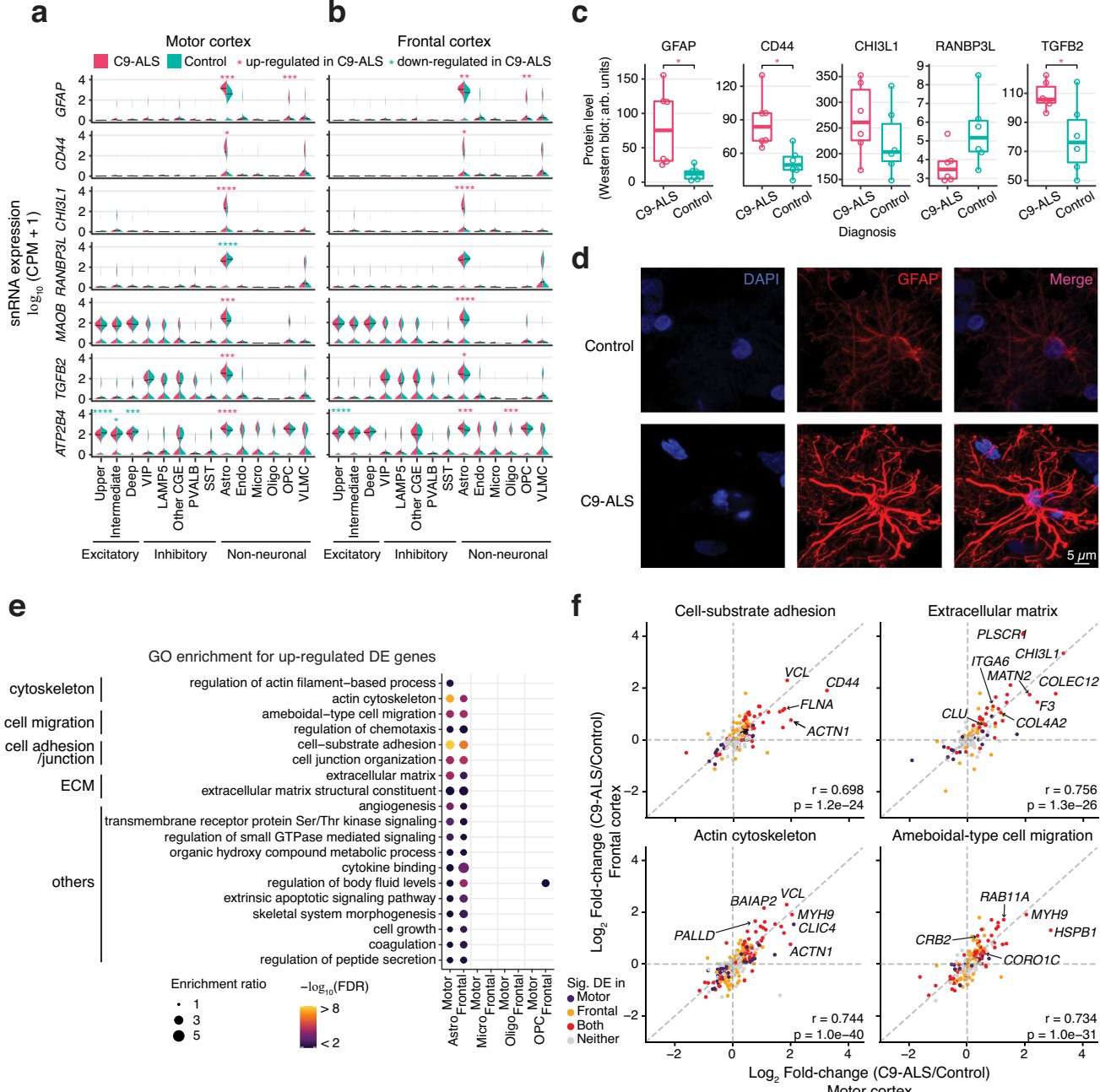

**Fig. 3 | Dysregulation of gene expression in C9-ALS astrocytes suggests activation and structural remodeling. a, b** Examples of genes that were prominently dysregulated in C9-ALS astrocytes. Asterisks marked the significant upregulated (pink) and downregulated (cyan) DE genes in C9-ALS from the MAST analysis. *FDR < 0.05; **FDR < 0.01; ***FDR < 0.001; ****FDR < 0.0001. **c** Examples of DE genes in astrocytes with corresponding changes in their protein products. Protein levels were measured by automated Western blot analysis. Arb. units, arbitrary units. N = 6 C9-ALS and 6 control biologically independent samples, and each sample was measured in two technical replicates. In each box plot, dots are the average from the two technical replicates for each sample; the lower and upper hinges correspond to the first and third quartiles; the whiskers extend 1.5 * IQR (interquartile range) away from the hinges; and the center denotes the median. *Two-sided Welch's t-test p = 0.033, 0.012, and 0.033 for GFAP, CD44 and TGFB2, respectively. See Supplementary Dataset 10 for raw values. **d** Representative

immunofluorescence images of the GFAP-positive astrocytes in motor cortex (red) obtained using confocal microscopy. DAPI was used to stain nuclei (blue). Immunofluorescence showed strong upregulation of GFAP immunoreactivity in astrocytes in C9-ALS donors. The images were acquired in 2 control and 2 C9-ALS subjects, 3 sections per subject and 3 fields in each section. **e** Top Gene Ontology (GO) terms enriched for upregulated DE genes in astrocytes. Enrichment of the same terms for upregulated DE genes in other glia cell types are shown as a comparison. Enrichment ratio is the number of observed genes divided by the number of expected genes from each GO term. ECM extracellular matrix. See Supplementary Dataset 5 for the full list of GO enrichment results. **f** Genes in four functional categories had consistent patterns of differential expression in astrocytes from both brain regions. r and p, two-sided Pearson correlation test coefficient and p-value. Source data are provided as a Source Data file.

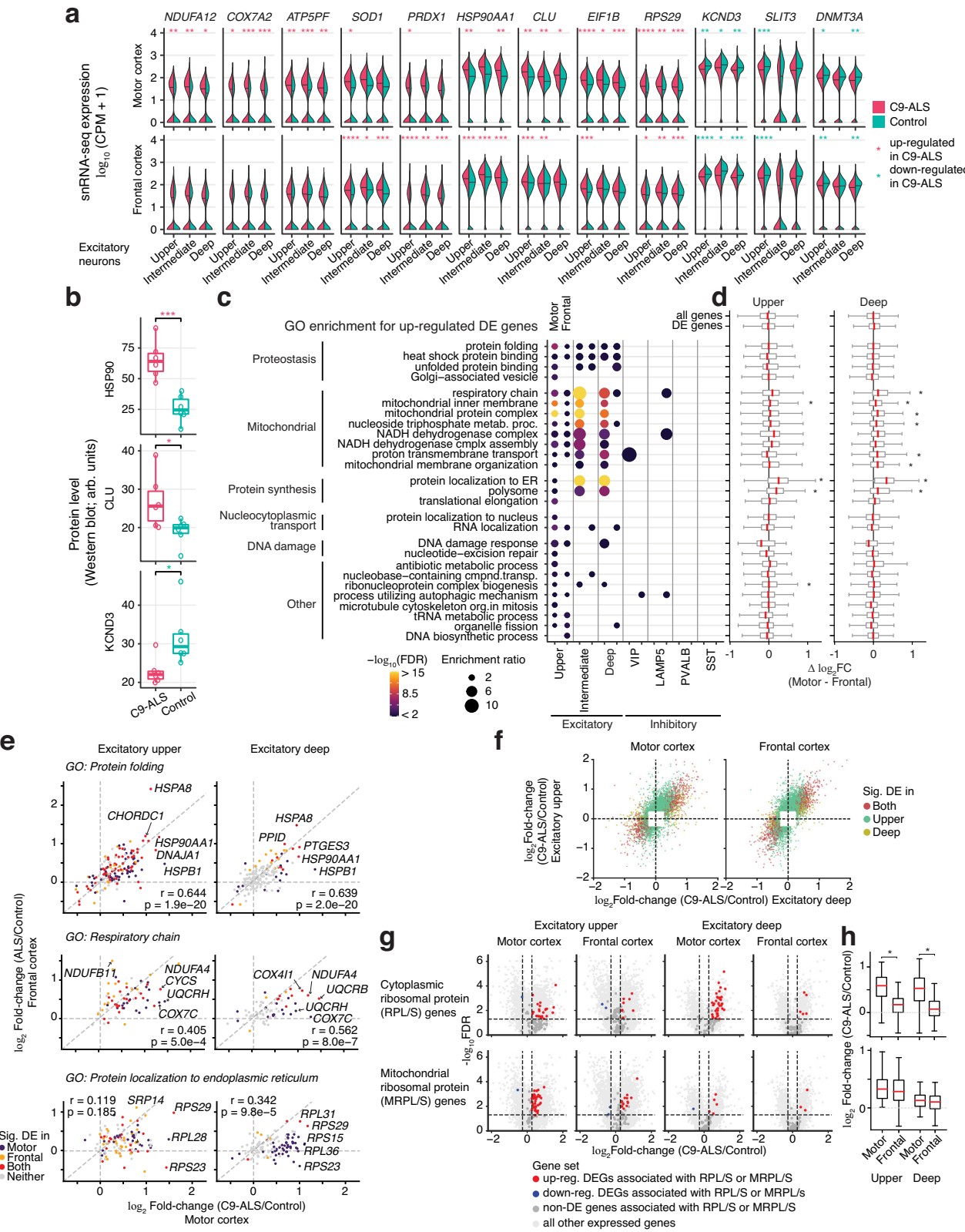

(Complexes I-IV), such as NADH ubiquinone oxidoreductase (*NDUF* family genes), succinate dehydrogenase (*SDHA, SDHB, SDHC*), cytochrome bc1 complex (*UQCR* family and *CYC1*), and cytochrome c oxidase (*COX* family and *CYCS*), as well as ATP synthase complex (*ATP5* family) (Fig. 4a; Supplementary Dataset 3, 4). Genes that encode proteins involved in mitochondrial membrane transport (*SLC25A* family, including *SLC25A4*) and chaperones that assist the import of proteins

from the cytoplasm into the mitochondrial inner membrane (*TIMM* family, e.g., *TIMMDC1*) were also upregulated.

Among upregulated genes associated with proteostasis, there was a large group of genes that encode members of the heat shock protein 70 (HSP70) molecular chaperone family (e.g., *HSPA4, HSPA8, HSPA9*) (Fig. 4e; Supplementary Dataset 3, 4)[47]. HSP70 proteins are required for aggregation prevention, folding of newly synthesized proteins,

**Fig. 4 | Excitatory neurons have altered expression of metabolic and protein regulatory pathway genes. a** Examples of genes that were dysregulated in C9-ALS excitatory neurons. Asterisks marked the upregulated and downregulated DE genes in C9-ALS from the MAST analysis. *FDR < 0.05; **FDR < 0.01; ***FDR < 0.001; ****FDR < 0.0001. CPM counts per million. **b** Examples of DE genes for which corresponding changes in protein abundance were confirmed by automated Western blot. Arb. units, arbitrary units. *N* = 6 C9-ALS and 6 control biologically independent samples, and each sample was measured in two technical replicates. Dots are the average from the two technical replicates for each sample. Asterisks denote two-sided Welch's *t*-test *p* = 6.35e-4, 0.041, and 0.036 for HSP90, CLU and KCND3, respectively. See Supplementary Dataset 10 for raw values. **c** Top GO terms enriched for genes upregulated in upper and/or deep layer excitatory neurons. Enrichments of the same terms for upregulated DE genes in other neuronal cell types are shown for comparison (see Supplementary Dataset 5). Enriched GO categories (FDR < 0.01) were selected by affinity propagation. Enrichment ratio is the number of observed genes divided by the number of expected genes from each GO term. **d** Differences in the C9-ALS vs. control fold-changes between motor and frontal cortex ($\Delta$ log$_2$FC = log$_2$FC in motor cortex−log$_2$FC in frontal cortex). The boxes denote the distribution of these differences ($\Delta$ log$_2$FC) of all expressed genes in each GO category. As background comparisons, the distributions of these differences for all expressed genes (labeled as "all genes") and for all C9-ALS vs. control DE genes (labeled as "DE gene") are shown on top. Two-sided Welch's *t*-tests

were used to test whether the $\Delta$ log$_2$FC in each group of genes were significantly different from the $\Delta$ log$_2$FC of the "all genes" control set, and * mark the significant differences (FDR < 0.005). Exact N numbers and *p*-values for this analysis are provided in Supplementary Dataset 5. **e** Comparison of effects in motor and frontal cortices for GO categories exemplifying three major cellular processes enriched for upregulated genes. r, Pearson correlation coefficient. **f** Comparison of the C9-ALS vs. control expression fold-changes between DE genes in upper- and deep-layer excitatory neurons. **g** Volcano plots demonstrating the dysregulation of genes associated with cytoplasmic ribosomal proteins (RPL/S, top row) and mitochondrial ribosomal proteins (MRPL/S, bottom row) in upper- and deep-layer excitatory neurons. Dots represent significant upregulated genes (red), downregulated genes (blue), and non-DE genes (dark gray) that are associated with cytoplasmic/mitochondrial ribosomal proteins, and all other expressed genes (light gray). **h** Comparison of the C9-ALS vs. control expression fold-changes of cytoplasmic ribosomal protein genes (*n* = 87, top row) and mitochondrial ribosomal protein genes (*n* = 72, bottom row) across the neuronal subtypes and cortical regions. *Two-sided Welch's *t*-test *p*-value 2.21e-15 and 1.38e-13 from left to right, respectively. In each box plot in panels (**b**), (**d**) and (**h**), the lower and upper hinges correspond to the first and third quartiles; the whiskers extend 1.5 * IQR (interquartile range) away from the hinges; and the center denotes the median. Source data are provided as a Source Data file.

conformational maintenance, as well as degradation of misfolded proteins and protein aggregates in autophagic-lysosomal or ubiquitin-proteasome pathways[48]. HSP70s carry out their functions together with members of large families of co-chaperones. Genes encoding multiple co-chaperones of HSP70 were also upregulated in C9-ALS excitatory neurons, such as members of *DNAJ* and *BAG* families (Supplementary Dataset 3, 4), as well as several other heat shock proteins (e.g., *HSPB1*, *HSPB11*) that facilitate the function of HSP70s in different protein quality control systems, including the degradation of protein aggregates in chaperone assisted selective autophagy[49–51]. Members of the HSP90 family of chaperons were also upregulated in C9-ALS excitatory neurons (Fig. 4a, e). Similar to HSP70, HSP90 chaperones bind to misfolded proteins, including TDP-43, and assist in their transition to native conformation or prevent their accumulation in misfolded state[52]. Of particular interest is the 1.8-fold increase in expression of *CLU* in upper layer excitatory neurons (Fig. 4a). Notably, upregulation of *CLU* was also detected in astrocytes (Fig. 3f). *CLU*, which is a risk gene for AD[53], encodes a normally secreted chaperone protein, clusterin, that is redirected to the cytosol during ER stress. Clusterin directly interacts with TDP-43 in vitro and potently inhibits its aggregation. Thus, increased expression of clusterin could provide an important defense against intracellular proteotoxicity associated with ALS[54]. We confirmed the upregulation of HSP90 and clusterin proteins in C9-ALS motor cortex (Fig. 4b; Supplementary Fig. 7).

Nearly half of the nucleus-encoded genes for cytoplasmic (*RPL/S* families) and mitochondrial (*MRPL/S* families) ribosomal proteins were upregulated in C9-ALS (Supplementary Dataset 3, 4). In both upper and deep excitatory neurons, we detected a larger number of upregulated cytoplasmic ribosomal genes in motor cortex compared to frontal cortex (Fig. 4g, h). In contrast, this difference was not observed for the mitochondrial ribosomal genes. Other C9-ALS upregulated genes that are associated with protein synthesis were those encoding translation initiation factors (e.g., *EIF1B*) (Fig. 4a). These intricate differences could result from region and/or cell specificity of the pathological changes in C9-ALS brain and will require further study.

Genes upregulated in C9-ALS excitatory neurons were linked to processes that are essential for all cell types (e.g., mitochondria function, proteostasis, protein synthesis), whereas genes downregulated in motor cortex were enriched for categories that are specific to neuronal cells (*e.g.*, neuronal cell body category that includes *C9orf72*, and genes that encode several potassium channels) (Fig. 4a; Supplementary Fig. 8; Supplementary Dataset 3, 4). Downregulated DE genes were also enriched for categories related to neuronal projections, and cell

adhesion. Similar to upregulated genes, in both cortical regions, the C9-ALS-associated downregulated genes were more strongly affected in upper *vs.* deep layer excitatory neurons (Fig. 4f). Likewise, we did not find significant differences in differential expression between motor and frontal cortices when we compared genes from the GO categories that were enriched in either cortical region (Supplementary Fig. 8a, b). We also examined the distribution of DE genes across fine-grained excitatory neuron cell types, and found broad enrichment across upper-layer neuron types as well as L6 intra-telencephalic (IT) neurons (Supplementary Fig. 8c).

The downregulated genes include genes that encode brain derived neurotrophic factor (*BDNF*), slit guidance ligands and their receptor (*SLIT1, SLIT3, ROBO2*), semaphorin proteins and their receptors (*SEMA4B, SEMA4C, SEMA5B, PLXNA1, PLXNA2, PLXNB2*), and RAP1 GTPase activating protein (*RAP1GAP*). All these proteins are linked to neuronal plasticity in the developing and/or adult brain, and *BDNF* has been implicated in the pathophysiology of several psychiatric and neurodegenerative diseases including ALS[55–60]. Also notable is down-regulation of genes that encode several potassium channels (e.g., *KCNN3, KCND3*)[61]. Unexpectedly, we observed ~45% decrease in expression of de novo DNA methyltransferase *DNMT3A* in the upper- and deep-layer excitatory neurons in motor cortex, suggesting a link to epigenetic dysregulation in C9-ALS. C9-ALS-associated changes in *DNMT3A* levels were previously reported in several studies, yielding conflicting results[62–64]. Consistent with our gene expression data, we detected downregulation of KCND3 protein, as well as a trend for RAP1GAP and DNMT3A proteins in C9-ALS motor cortex (Fig. 4b; Supplementary Fig. 7).

## Disruption of epigenetic regulation in C9-ALS neurons and glial cells

The widespread alterations of gene expression in C9-ALS excitatory neurons, astrocytes, and other cell types raises the question of how such transcriptional changes are established and maintained. We used two complementary assays to determine the epigenetic landscape of C9-ALS brain cells. Single nucleus ATAC-Seq (snATAC-seq) identifies regions of accessible chromatin in single cells[65,66]. We generated 109,198 high-quality snATAC-seq profiles (TSS enrichment ≥ 4, unique fragments ≥ 1000 per cell, Supplementary Fig. 9), and clustered these to generate pseudo-bulk accessibility profiles for 11 major brain cell types (Supplementary Fig. 10a). The snATAC clusters were annotated by transferring labels from the snRNA-seq data (Supplementary Dataset 6; Methods). The clusters were not driven by brain region, donor

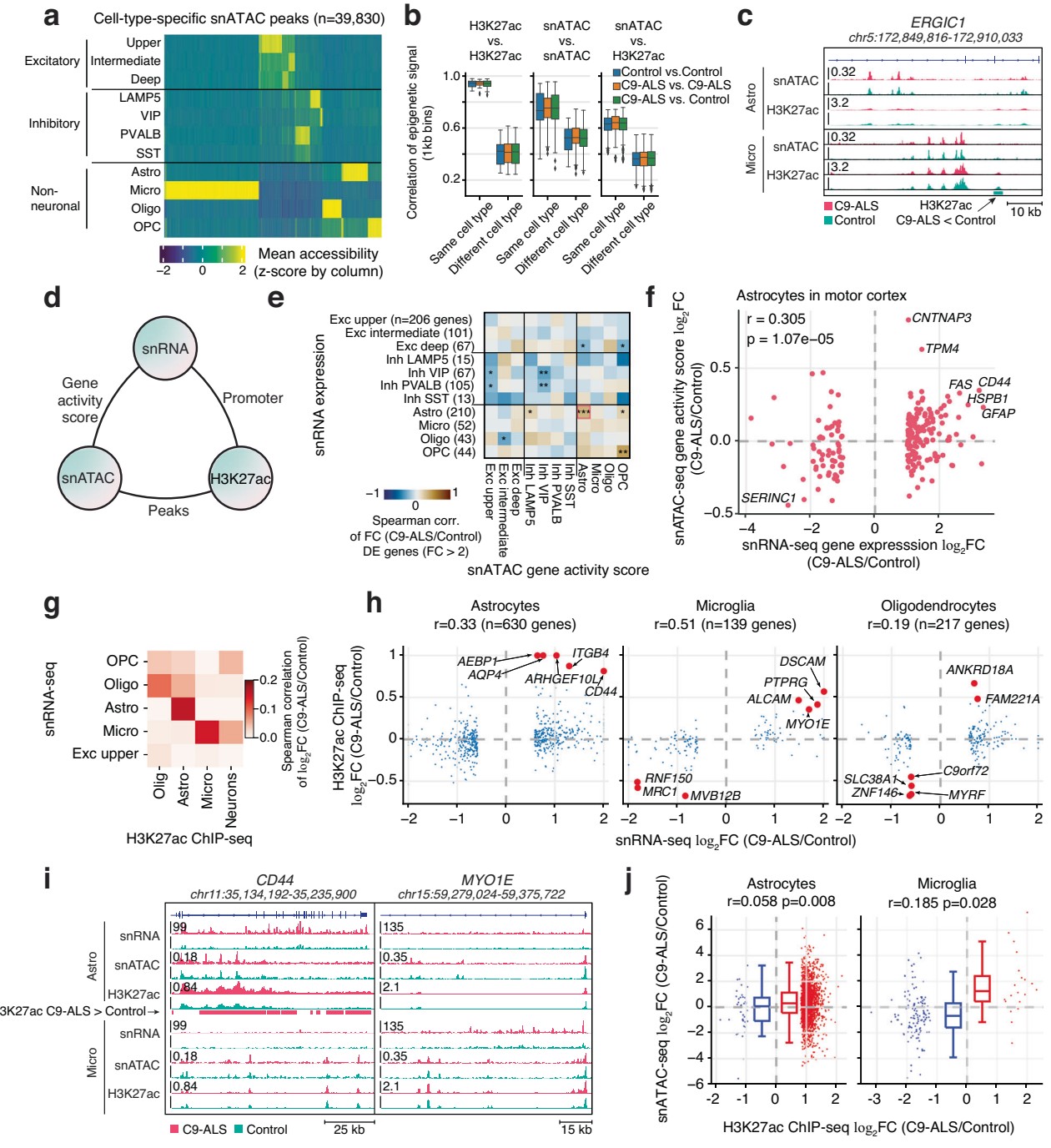

sex, individual samples, or diagnosis (Supplementary Fig. 10b), and they had consistent patterns of accessibility at cell type marker genes (Supplementary Fig. 10c). Similar to our snRNA-seq data, we observed fewer nuclei that passed QC in C9-FTD, and C9-FTD nuclei that passed QC also generally had lower TSS enrichment score and number of unique fragments in all major cell types (Supplementary Fig. 10d, e).

We found 39,830 cell-type-specific snATAC peaks marking regions of accessible chromatin (FDR < 0.05, $\log_2$FC > 0.5 in one cell type vs. the rest; Fig. 5a). However, we identified no significant (FDR < 0.05) differential peaks between C9-ALS and control in each major cell type. We also estimated the chromatin accessibility within peaks sharing the same transcription factor (TF) motif with ChromVAR[67] (Supplementary Fig. 11). Again, we observed that the variability of ChromVAR scores across nuclei for TF motifs were mainly

contributed by the cell type differences. For example, SOX, SPI and TCF were among the topmost variable TF motifs in terms of Chrom-VAR scores (Supplementary Fig. 11a), and their accessibility were enriched in oligodendrocytes, microglia, and neurons, respectively (Supplementary Fig. 11b–d).

Although snATAC-seq can in principle provide fine-grained resolution of cell types, the accuracy of clustering and thus the reliability of the pseudo-bulk profiles is limited by the sparse data from individual cells. Indeed, we found that the distinction between neuronal cell types was not as clear using snATAC-seq data (Supplementary Fig. 10a, c) as it was using snRNA-seq (Fig. 1a). We therefore used a complementary strategy, isolating bulk samples of nuclei from four major populations by FANS. Using antibodies against three cell-type-specific nuclear markers, we purified astrocytes, oligodendrocyte lineage cells,

**Fig. 5 | Epigenetic alterations correlate with transcriptome dysregulation in C9-ALS. a** Normalized chromatin accessibility at cell-type-specific snATAC peaks in 11 major cell types identified in snATAC-seq. **b** Pearson correlation of snATAC and H3K27ac ChIP-seq signal in 1 kb genomic bins. Box-and-whisker plots show the distribution of correlations between replicates for the same cell type, or across different cell types. N numbers for each group are provided in Supplementary Dataset 11. **c** Genome browser view of snATAC-seq and H3K27ac ChIP-seq signals from astrocytes and microglia at the *ERGIC1* locus. Track height represents pseudo-bulk counts normalized by reads in TSS for snATAC-seq, or average signal (counts normalized to one million reads) across donors for ChIP-seq. Bottom track highlights a H3K27ac peak that is significantly reduced in C9-ALS. **d** Schematic of pairwise comparisons (shown in subsequent panels) of C9-ALS effects on the transcriptome (snRNA-seq), chromatin accessibility (snATAC-seq), and histone modification H3K27ac (ChIP-Seq). For each pair of data modalities, we correlated the fold-change of gene- or H3K27ac peak-associated signals. **e, f** Comparison of snRNA with snATAC. **e** Spearman correlation of the C9-ALS vs. control fold-change (FC) for snRNA expression vs. snATAC gene activity score in motor cortex. The analysis was limited to strongly DE genes (FC > 2) in each major cell type in motor cortex. Two-sided Spearman's rank correlation test: *$p < 0.05$; **$p < 0.01$; ***$p < 0.001$. Exact values of *r* and *p* are provided in Supplementary Dataset 12. See Supplementary Fig. 10f for frontal cortex data. **f** Scatter plot illustrating the significant correlation between differential gene expression (snRNA-seq) and snATAC-seq changes in astrocytes in motor cortex. Selected genes with high concordant FC are labeled. *r* and *p*, two-sided Spearman's rank correlation test coefficient and *p*-value. **g, h** Comparison of snRNA with H3K27ac ChIP-Seq, showing Spearman correlation (**g**) and scatter plots (**h**) of DE gene expression (snRNA-seq) vs. promoter H3K27ac signal (ChIP-seq). Genes with concordant and biggest H3K27ac signal fold-changes were highlighted in red. **i** Browser view of the *CD44* and *MYO1E* loci, showing the correspondence of epigenomic and transcriptomic signals. Track height represents average RPKM across donors for snRNA-seq, pseudo-bulk counts normalized by reads in TSS for snATAC-seq, or average signal (counts normalized to one million reads) across donors for ChIP-seq. Pink rectangles highlight significant H3K27ac differential peaks that were increased in astrocytes from C9-ALS samples. **j** Correlation of snATAC vs. H3K27ac ChIP-seq signal at differential H3K27ac peaks. Box plots show the distribution for upregulated ($N = 2054$ for astrocytes and 20 for microglia) and downregulated peaks ($N = 40$ for astrocytes and 122 for microglia). *r* and *p*, two-sided Spearman's rank correlation test coefficient and *p*-value. In each box plot in panels (**b**) and (**j**), the lower and upper hinges correspond to the first and third quartiles; the whiskers extend 1.5 * IQR (interquartile range) away from the hinges; and the center denotes the median. Source data are provided as a Source Data file.

microglia, and neurons (Supplementary Fig. 5b). We then profiled active enhancers and promoters marked by histone 3 lysine 27 acetylation (H3K27ac) using ChIP-seq (Supplementary Fig. 5c)[68]. The ChIP-seq data had very high signal/noise ratio, with strong correlation between replicates for the same cell type ($r = 0.94 \pm 0.029$, mean ± s.d.) and lower correlation for different cell types ($r = 0.41 \pm 0.084$; Fig. 5b; Supplementary Dataset 11). The correlation was equally strong between C9-ALS and control subjects as it was between control subjects. snATAC data had lower inter-replicate correlation for the same cell type ($r = 0.76 \pm 0.16$) and higher correlation between cell types ($r = 0.52 \pm 0.11$; Fig. 5b; Supplementary Dataset 11).

We observed that the ChIP-seq data correlated strongly with the snATAC-seq tracks from corresponding cell types ($r = 0.62 \pm 0.086$), while the correlation for different cell types was lower ($r = 0.36 \pm 0.087$; Fig. 5b; Supplementary Dataset 11). Both assays marked cell-type-specific regions of accessible and active regulatory chromatin. The correspondence between H3K27ac and snATAC data at cell-type-specific enhancers was evident at the locus of the ALS risk gene *ERGIC1*[69], where multiple snATAC peaks coincide with H3K27ac peaks (Fig. 5c). The ChIP-seq tracks generated from bulk nuclei had deeper coverage than the snATAC-seq data (on average 2.25-fold more unique mapped sequence fragments), and we observed a corresponding higher signal/noise ratio in the H3K27ac tracks (Fig. 5c).

We found multiple differentially acetylated (DA) chromatin regions in C9-ALS subjects compared with controls (FDR < 0.05; Supplementary Dataset 7). There was a large difference in numbers of DA regions between the cell types, with 16 DA regions detected in neurons vs. 2945 in astrocytes. For example, we observed more active chromatin (increased H3K27ac), along with increased gene expression, at the *CD44* locus in astrocytes from C9-ALS compared to control subjects (Fig. 5i). This disparity is likely explained by heterogeneity of neuronal subtypes which precludes the detection of the DA regions in the bulk population. Notably, as stated above, differential accessibility analysis of the snATAC-seq datasets did not reveal significant differences between ALS and control subjects even in astrocytes. We attributed the lack of differential snATAC-seq peaks to the lower signal/noise ratio in these datasets, as well as the statistical burden of multiple comparisons in analysis of the thousands of genome-wide open chromatin regions.

To further characterize patterns of disease-related differences in epigenetic regulation and connect them with transcriptional alterations, we investigated genome-wide correlations among all three data types: H3K27ac ChIP-Seq, snATAC-seq, and snRNA-seq. In particular, we examined the relationship between cell-type-specific signatures of

ALS in each pair of datasets using gene expression, gene activity score, and H3K27ac peaks (Fig. 5d). To link gene expression with open chromatin (snRNA vs. snATAC), we calculated the gene activity score for strongly DE genes (FC > 2) (Methods). Disease-associated changes in gene expression correlated with open chromatin for astrocytes and OPCs in motor cortex, and in astrocytes and microglia in frontal cortex (FDR < 0.05, Fig. 5e, Supplementary Fig. 10f; Supplementary Dataset 12). Indeed, for astrocytes we found that key upregulated markers including *GFAP, CD44, HSPB1, TPM4*, and *CNTNAP3*, contained more open chromatin in C9-ALS subjects (Fig. 5f). By contrast, significantly downregulated genes such as *SERINC1* contained less open chromatin. Notably, we did not observe a significant correlation between snATAC and snRNA signatures of C9-ALS neurons, oligodendrocytes, or microglia, potentially indicating that chromatin accessibility is less affected in those cell types.

Next, we compared differential expression with chromatin activation at the gene promoter (snRNA vs. H3K27ac). This analysis revealed a strong positive correlation for three glial cell types (astrocytes, oligodendrocytes, and microglia; $p < 10^{-5}$, Fig. 5g, h). The correlation was especially strong for microglia, even though fewer genes were DE in microglia compared with astrocytes ($r = 0.51$ for microglia and $r = 0.33$ for astrocytes). The strong consistency of C9-ALS-related differential snRNA and H3K27ac signals in microglia supports the robustness of our observations in these cells. For example, we found that the gene *MYO1E* was transcriptionally upregulated in microglia and had a corresponding enrichment of active chromatin at its promoter (Fig. 5h, i). Whereas ALS-related differences in expression were correlated with H3K27ac in a cell-type-specific manner for the non-neuronal cells (Fig. 5g), the neuronal ChIP-seq data was less consistent with snRNA, most likely because it represents a mixture of diverse neuronal subtypes.

Finally, we directly compared the two epigenetic data types for the glial cells (Fig. 5j), focusing on H3K27ac differentially acetylated peaks. We found that C9-ALS-related changes in chromatin accessibility were positively correlated with H3K27ac in both astrocytes and microglia ($p < 0.05$, Fig. 5j).

## A distinct signature of glial dysregulation in C9-FTD

Our initial analyses showed striking differences in snRNA-seq data from C9-FTD compared with C9-ALS, despite the shared genetic risk factor. Notably, the proportion of high-quality excitatory and inhibitory neurons was 2.1-fold lower in frontal cortex in C9-FTD compared with control (Fig. 6a, b). No such reduction in the proportion of neurons was observed in C9-ALS. The reduction in high-quality neurons in

C9-FTD samples was mainly due to a high proportion of low-quality nuclei with a smaller number of detected genes and a high proportion of mitochondrial RNA (Supplementary Fig. 1a–c; Supplementary Fig. 12a). Notably, flow cytometry showed no difference in the overall proportion of neurons (NeuN+ SOX10- cells) in any of the donor groups (Supplementary Fig. 5d), suggesting that the depletion of the frontal cortex C9-FTD neurons in snRNA-seq was due to the lower quality of neuronal nuclear RNA. Neurons were also depleted among high-quality snRNA-seq nuclei in many of the motor cortical samples, but the proportion was more variable and not significantly different from controls on average (Fig. 6b). Among neurons, the ratio of excitatory to inhibitory neurons was not significantly different (Fig. 6c; Supplementary Fig. 12b). Although our study cannot distinguish between lower tissue quality due to technical factors (e.g., tissue handling) vs. disease effects, the lower proportion of high-quality nuclei that we observed was limited to a specific cell type (neurons), donor population (C9-FTD), and brain region (frontal cortex), which is consistent with the expected pattern of tissue-level pathology in C9-FTD.

The number of high-quality neurons sampled from C9-FTD patients was too low for statistically meaningful analysis of the transcriptome. However, our data enabled high-resolution analysis of the molecular dysregulation in C9-FTD glial cells. We found thousands of DE genes in astrocytes, oligodendrocytes and OPCs, as well as a smaller but significant number of DE genes in microglia (Fig. 6d; Supplementary Dataset 8, 9). After downsampling to equalize the statistical power across non-neuronal cell types and brain regions, we found that astrocytes and OPCs were the most strongly affected cell types (Fig. 6e). We were surprised to find more DE genes in glial cells in motor cortex than in frontal cortex, in contrast to the more pronounced depletion of high quality neurons in frontal cortex (Fig. 6e). This observation could indicate that glial responses to neuronal degeneration in C9-FTD are distributed through multiple cortical regions. In both motor and frontal cortices, we found strongly correlated differential gene expression fold-changes in the single nucleus and FANS RNA-seq data for corresponding glial cell types (Fig. 6f). As in C9-ALS, we found that glial DE genes in C9-FTD had consistent effect sizes in both cortical regions (Fig. 6g). We clustered the glial DE genes in C9-FTD, identifying groups of genes specifically affected in oligodendrocytes and OPCs (C2-C5) or in astrocytes (C6, C7) (Fig. 6h). These genes exhibited generally consistent effects in both motor and frontal cortices.

We next directly compared the disease effects in glia cells between C9-ALS and C9-FTD. To avoid double dipping, we split the 6 control donors into two groups and ran a second differential test for each glial cell type (Methods). We detected only a small number of shared DE genes between the two diseases (Supplementary Fig. 12c). Also, the most strongly affected DE genes (fold-change > 2) in one disease showed inconsistent disease-associated changes in the other disease (Supplementary Fig. 12d–e). To identify the most pronounced differences, we looked at genes that were DE in both C9-ALS and C9-FTD, and highlighted several genes that were altered in the same or in opposite directions (Fig. 6i). We found that *GFAP* was strongly upregulated in motor cortex in both diseases, consistent with activation of astrocytes. On the other hand, the voltage-gated potassium channel regulator protein, *LGI1*, was upregulated in C9-FTD but downregulated in C9-ALS in astrocytes from motor cortex[70]. Notably, clearance of extracellular potassium by astrocytes is essential for controlling neuronal excitability[71]. *NAV2* that encodes Neuron Navigator 2, showed an opposite pattern of expression in oligodendrocytes from frontal cortex. Neuron Navigator 2 was suggested to be involved in CNS development, and the *NAV2* gene has been implicated in Alzheimer's disease[72,73]. These results suggest a distinct pattern of transcriptome dysregulation in C9-ALS and C9-FTD.

## Discussion

Both sporadic and C9-associated ALS and FTD are widely considered to be initiated by degenerative processes in motor or von Economo neurons, respectively[16–18]. Yet, mounting evidence indicates that the broader population of neuronal and non-neuronal cells play key roles in the pathology of these diseases. Here, we used single nucleus transcriptome and epigenome sequencing from autopsied human motor and frontal cortices of cases and controls to identify the specific cell types and the biological pathways that are altered in C9-ALS or C9-FTD. These data, together with bulk RNA-seq and epigenetic (H3K27ac ChIP-seq) assays in FANS-separated nuclei, revealed distinct molecular pathologies in C9-ALS and C9-FTD across cell types and brain regions.

Previous studies have implicated all major classes of glial cells and blood-borne immune cells in the pathology of ALS[22,74–77]. Yet, it remains unclear how these findings, largely based on animal or in vitro models of (C9)-ALS, may translate to human ALS patients. Our analysis showed that among non-neuronal cells, astrocytes were the most strongly affected in C9-ALS. We found comparatively fewer gene expression changes in microglia and oligodendrocytes. This finding may be partly explained by the fact that our data reflect the end-of-life transcriptional disruption in C9-ALS, rather than gene expression changes at earlier stages of the disease. We detected transcriptional upregulation of markers of astrocyte activation, and of cytoskeleton and cell-surface protein remodeling in C9-ALS astrocytes. This is consistent with the hypothesis of a common astrocytic response to many forms of injury and neurodegenerative diseases, termed 'astrogliosis' or 'astrocyte reactivity', which are often accompanied by profound structural changes, including changes in different filament systems and the actin cytoskeleton[32]. Similar to C9-ALS, C9-FTD astrocytes also exhibited upregulation of activation markers (*GFAP, CD44*).

Anatomical studies showed that ALS causes degeneration of the giant upper motor neurons (Betz cells) which are rare cells located in layer 5b of the primary motor cortex and account for only ~10% of the pyramidal cells in this layer[25]. A recent large-scale study of the human motor cortex identified cells consistent with the size and shape of Betz cells in two Layer 5 extratelencephalic clusters (Exc L3-5 *FEZF2 ASGR2* and Exc L5 *FEZF2 CSN1S1*), but these clusters also included other neurons with pyramidal morphologies[78]. In the absence of specific markers, we were not able to reliably separate a cluster corresponding to Betz cells in our snRNA-seq data. However, all deep layer (L5/6) excitatory neurons are generally considered vulnerable in ALS[21,79], and we found transcriptional dysregulation of L5/6 excitatory neurons which was more extensive in motor than in frontal cortex. Surprisingly, we found even more extensive gene dysregulation in C9-ALS in upper layer (L2/3) excitatory neurons, which were equally afflicted in both cortical regions. This finding is intriguing given that L2/3 neurons, which mainly project intratelencephalically to connect to distant cortical regions and striatum, are greatly expanded as a proportion of the cortical neuron population in human compared with marmoset or mouse[24,80]. The extensive pathological alterations in L2/3 neurons in both motor and frontal cortices could thus contribute to the cognitive and behavioral symptoms in C9-ALS[81].

Whereas genes downregulated in C9-ALS excitatory neurons showed the most significant enrichments for categories associated with neuron-specific functions, probably reflecting neuronal degeneration associated with the disease, the upregulated genes mostly affected mitochondrial function, proteostasis, and protein synthesis pathways. Impaired mitochondrial energy production and functions (e.g., maintenance of $Ca^{2+}$ homeostasis) have been described in many animal and in vitro models of ALS, albeit with conflicting results[82]. Our data showed upregulation of the nucleus-encoded mitochondrial genes in deep and upper layer excitatory neurons in both cortical regions. By contrast, recent electron microscopy observations in autopsied motor cortex show decreased mitochondrial density and

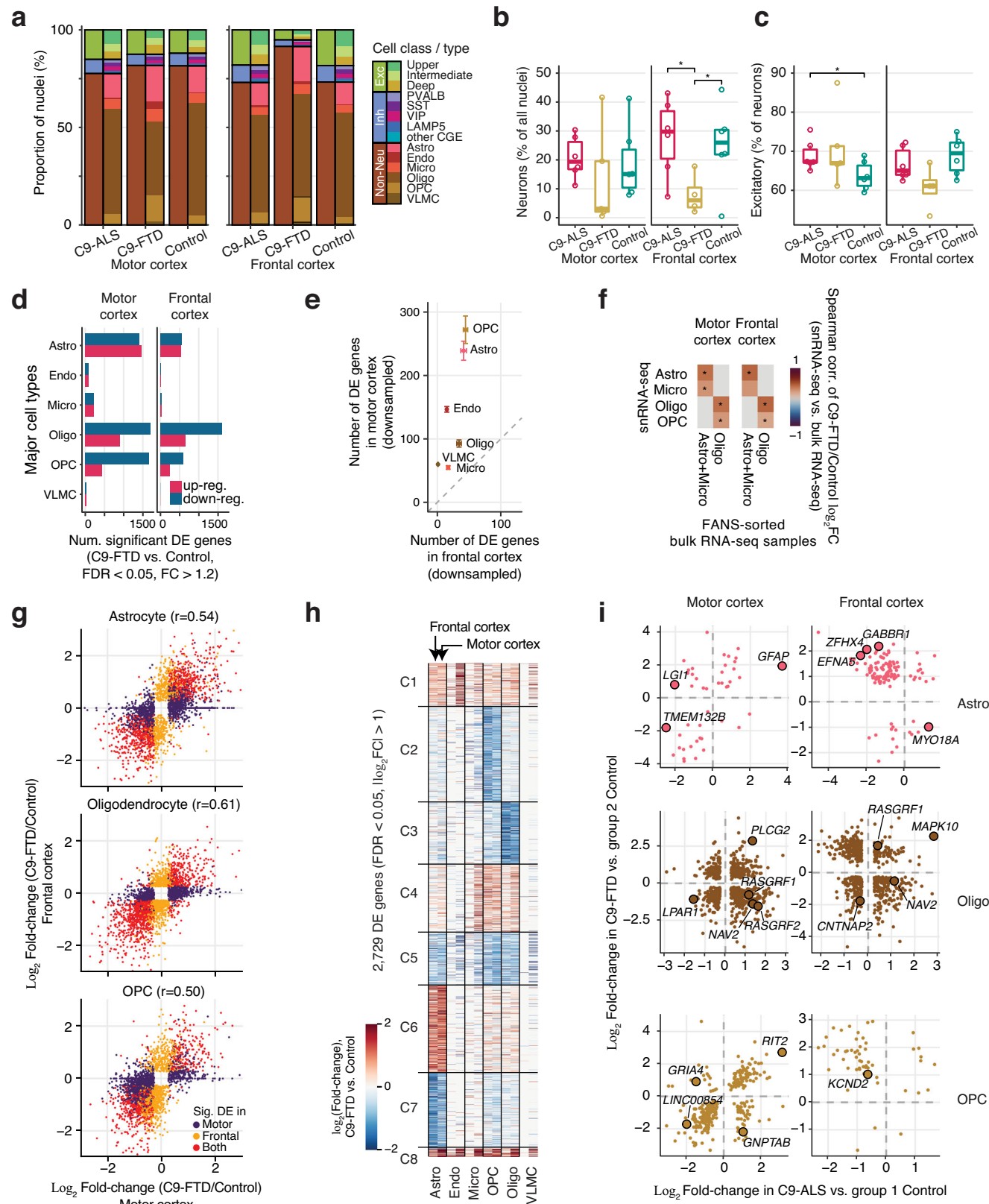

length in ALS patients[83]. Mitochondria were also decreased in animal and cell models of ALS[84]. We detected no changes for the three mitochondrial proteins that were encoded by the C9-ALS vs. controls DE genes and that were assessed in our Western blot experiments. Of note, mitochondrial genes are highly expressed in all cell types and our protein assay was performed in bulk brain tissue, precluding the detection of the neuron-specific signal. Therefore, we cannot reliably evaluate the relationship between these transcriptional and protein alterations in the C9-ALS neurons. Still, our findings suggest that upregulation of mitochondrial genes in C9-ALS could reflect adaptation in the surviving mitochondria, at least at the transcriptional level, potentially to satisfy the energy demands of the neuronal cells. It has

**Fig. 6 | Lower proportion of high-quality neurons and alteration of non-neuronal transcription in C9-FTD. a** Distribution of the abundance of three cell classes and 14 major cell types across brain regions and diagnoses. **b, c** Relative abundance of neurons (**b**), and percent of excitatory neurons among all neurons (**c**). Circles represent biologically independent individual donors; $N = 6$ C9-ALS and 6 control samples for both brain regions, and $N = 5$ and 4 C9-FTD samples from motor cortex and frontal cortex, respectively. In each box plot, the lower and upper hinges correspond to the first and third quartiles; the whiskers extend 1.5 * IQR (interquartile range) away from the hinges; and the center denotes the median. *Two-sided Welch's t-test $p = 0.016$, 0.041, and 0.039 for the comparisons from left to right, respectively. **d** The number of DE genes in FTD vs. control (FDR < 0.05, FC > 1.2) in glia. **e** Comparison of the number of FTD vs. control DE genes between the motor and frontal cortex after downsampling to 30 nuclei per donor in each cell type. Random downsampling was performed 10 times. Dots and error bars

represent mean ± SEM. **f** Spearman correlation between gene expression FC from snRNA-seq vs. FANS-sorted bulk RNA-seq. Correlation was performed using significant DE genes in C9-FTD vs. control identified in snRNA (FDR < 0.05, FC > 1.2). *Spearman correlation test, FDR < 0.05. **g** Transcriptional changes in C9-FTD were consistent in two cortical regions. r, Pearson correlation. **h** K-mean clustering analysis of strongly DE genes in C9-FTD vs. control (>2 fold-change). **i** Comparison of the disease fold-changes in C9-ALS and C9-FTD for astrocytes (Astro), oligodendrocytes (Oligo), and OPC. To avoid double-dipping, the control donors were split into two groups and used to compute the fold-changes for C9-ALS and C9-FTD respectively (see Methods). Only genes that were significantly DE in both comparisons with these split controls are shown in the scatter plot. Highlighted are examples of genes that are also significantly DE in our full model reported in Fig. 2a and Fig. 6d. Source data are provided as a Source Data file.

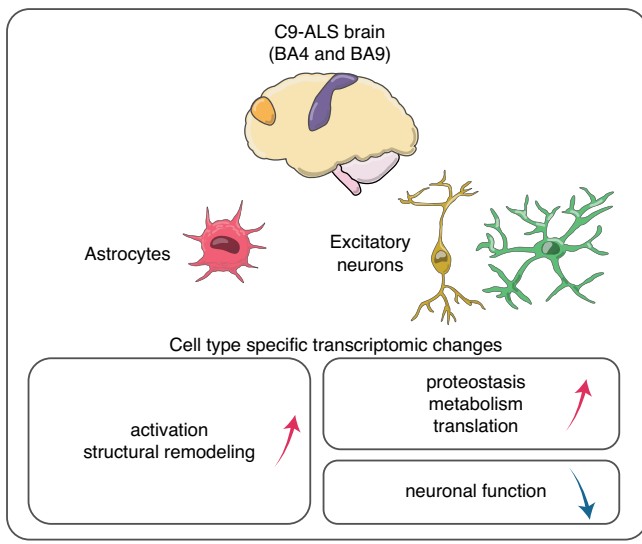

**Fig. 7 | Summary of the major results of our snRNA-seq analysis of postmortem brain samples from C9-ALS donors.** We detected that the most pronounced transcriptional disruption in C9-ALS concentrated in excitatory neurons and astrocytes. These changes were highly consistent in motor and frontal cortices. C9-ALS astrocytes showed increased expression of genes associated with activation and structural remodeling. C9-ALS upper-layer (L2/3) and deep-layer (L5/6) excitatory neurons had increased expression of genes related to proteostasis, metabolism, whereas genes related to neuronal function were downregulated. In both cortical regions, there were more extensive gene dysregulation in upper layer vs. deep-layer excitatory neurons.

been suggested that energy demand might be substantially higher in C9-ALS than healthy brains, as the accumulation of DPRs and TDP43 protein aggregates or their pathogenic oligomers in C9-ALS neurons triggers upregulation of multiple systems that maintain cellular homeostasis[85]. In turn, the high energy cost of homeostasis might require an unsustainable increase in energy production. Consistent with this connection, in addition to upregulation of mitochondrial genes, we detected upregulation of proteostasis genes that encode heat shock proteins and their co-chaperones, and we confirmed upregulation of several proteostasis proteins encoded by the DE genes in C9-ALS motor cortex. Our major snRNA-seq findings in C9-ALS are summarized in Fig. 7. Our global analyses of gene expression differences across donors showed particularly pronounced disruption of transcriptome in excitatory neurons in C9-ALS and astrocytes in both C9-FTD (Supplementary Figs. 14, 15).

Our epigenomic data, including single nucleus open chromatin (snATAC-seq) and histone modification H3K27ac chromatin immunoprecipitation (ChIP-seq), complement our cell type-specific transcriptome findings. The high quality of our epigenomic data from

multiple donors allowed us to identify significant C9-ALS disease-associated changes of H3K27ac at promoters and distal regulatory elements, which correlated with differential gene expression and chromatin accessibility in specific glial cell types. We further found coordinated effects in C9-ALS on a large number of distal regions marked by open chromatin and H3K27ac. Notably, epigenomic alterations were more pronounced in glial cells compared with neurons, although this could partly reflect the difficulty of distinguishing cells from diverse neuronal types in sparse snATAC-seq data.

Our study provides a high-resolution view of the transcriptional and epigenetic alterations in individual cell types from the brains of C9-ALS and C9-FTD donors and offers a valuable resource for the scientific community. A limitation of this approach is that we cannot distinguish cell-type-specific alterations that represent direct effects of C9 mutation from end-of-life consequences of neurodegeneration and the resulting disease which are present in the C9-ALS or C9-FTD cases. Also, our methods do not allow for a spatial resolution of the ALS-associated transcriptome dysregulation, which provides valuable information of how this dysregulation participates in the spread of ALS or FTD pathology[86]. Although we confirmed that differential expression of some genes results in the corresponding changes in protein expression, it is possible that some disease-associated mRNA effects are counteracted by homeostatic regulation of translation, justifying future large-scale proteomic studies of ALS and FTD brain. Lastly, a large number of low-quality neurons in C9-FTD patients precluded statistically meaningful analysis of the neuronal transcriptome. Overall, this work demonstrates the value of large-scale single cell/nucleus studies in patients to establish the cell-type-specific molecular pathology of C9-ALS and C9-FTD, which will be essential for developing targeted disease-modifying therapies.

## Methods
### Human samples
Human frozen postmortem tissues were obtained from the Mayo Clinic Brain Bank (Jacksonville, FL USA). Notably, 3 of the 12 control brains were obtained via the Einstein Aging Study. All brains were acquired with informed consent, and procedures were conducted according to the approved Institutional Review Board protocol (IRB# 15.009452). Data analysis was conducted as exempt human research, considering that these were postmortem brain samples, and they were not specifically collected for this study. The samples were dissected from the dorsolateral prefrontal cortex (Brodmann area 9; BA9; hereafter "frontal cortex") and/or motor cortex (Brodmann area 4; BA4), which are of particular relevance for ALS and FTD pathological diagnosis, respectively. A total of 17 donors with C9-ALS ($N = 6$), C9-FTD ($N = 5$), or controls determined to have no remarkable pathology ($N = 6$) were used for snRNA-seq, snATAC-seq, bulk RNA-seq in FANS-separated nuclei from major brain cell type, and automated Western blotting studies (Supplementary Dataset 1). A separate group of C9-ALS ($N = 6$) and normal (no remarkable

pathology, $N = 9$) donors was used in H3K27ac ChIP-seq studies (Supplementary Dataset 1).

All subjects were confirmed negative for protein-coding mutations in *TARDBP*, *FUS*, *NEK1*, *GRN*, *MAPT*, or *TBK1*. *C9orf72* repeat expansions were confirmed via repeat-primed PCR and Southern blotting. ALS/FTD neuropathologic changes were assessed in both BA4 and BA9 using immunohistochemistry for TAR DNA binding protein 43 (TDP-43). C9-FTD cases presented TDP pathology in PFC with almost no inclusions in motor cortex, along with no Betz cells depletion. In C9-ALS cases, TDP pathology was concentrated in motor cortex with a severe depletion of Betz cells. All C9-ALS cases had clinical features of muscle weakness and atrophy with fasciculations, with at most mild cognitive and memory impairment (3 of 10 cases) or mild dementia (2 of 10 cases) (Supplementary Dataset 1). None of the C9-ALS cases had parkinsonism or significant apathy. Only 1 case had mild depression and 2 cases had some personality changes. In contrast, all 7 FTD cases had dementia with behavioral abnormalities (most often frontal type), but only 2 had mild muscle weakness.

## Statistics and reproducibility

This study was designed to determine transcriptional and epigenetic alterations in the brains of ALS and FTD donors with the repeat expansion in *C9orf72* (C9) gene in single cell resolution. For this purpose, we performed snRNA-seq and snATAC-seq in postmortem samples from C9-ALS ($N = 6$), C9-FTD ($N = 5$) and pathologically normal donors ($N = 6$). To compare the impact across different cortical regions, these studies were performed in motor cortex and frontal cortex, which have been considered to be afflicted the most in ALS and FTD, respectively. For validation, the single nucleus studies were complemented by RNA-seq and epigenetic (H3K27ac ChIP-seq) assays in FANS-separated nuclei from major brain cell types, as well as by automated Western blotting in bulk motor cortex tissue. No statistical methods were used to predetermine sample size because effect sizes were unknown before experiments. The investigators were not blinded to the disease status of the donors. Samples were randomly assigned to batches for nuclear isolation and library preparation. All statistics tests were performed in R (v4.1.2) and Python (v3.7.12), and the details of the tests used in each analysis are provided in the main text, figure legends and the following paragraphs of the Method section.

## Nuclei isolation and snRNA/snATAC-seq using the 10X Genomics platform

Nuclei were isolated from 50–70 mg frozen brain tissue, using the previously published protocol[87]. For each donor, brain tissues from the motor cortex and frontal cortex were dissected with a scalpel blade. The tissues were then homogenized on ice in 5 mL lysis buffer (0.32 M sucrose, 3 mM Mg(Ac)$_2$, 3 mM CaCl$_2$, 0.1 mM EDTA, 1 mM DTT, 0.1% Triton-X, and 10 mM Tris-HCl pH 8.0 in DEPC-treated water), containing 0.4 U/μl freshly added Recombinant RNase Inhibitor (RRI; Takara, Cat. # 2313 A) until no chunks of tissue were visible in the tissue suspension (~20 strokes) using a glass Dounce homogenizer (Thomas Scientific; Cat. # 3431D76; size A). The homogenized tissues were filtered through a 40 μm strainer (Thermofisher; Cat. # 22-363-547) and transferred to a 30 mL thick polycarbonate ultracentrifuge tube (Beckman Coulter; Cat. # 355631). Nine mL of the sucrose buffer (1.8 M Sucrose, 3 mM Mg(Ac)$_2$, 1 mM DTT, and 10 mM Tris-HCl pH 8.0 in DEPC-treated water) were added to the bottom of the tubes with the tissue homogenate, and the tubes were centrifuged at 112,500 g (25,000 rpm using SW28 rotor) for 2.5 h at 4 °C. The supernatant was aspirated, and the nuclear pellet was submerged on ice for 20 min in 250 uL of DEPC-treated water-based PBS, containing 1% BSA and 0.4 U/μL RRI. The nuclear pellet was then resuspended and filtered twice through a 30 μm cell strainer (Miltenyi Pre-separation Filters; Cat. # 130-041-407). Nuclei were counted using a hemocytometer and diluted to ~2000 nuclei/μL before performing single-nucleus capture

on the 10X Genomics Single-Cell 3′ system. The 10X capture and library preparation protocols were used without modification. Matched control, C9-ALS and C9-FTD samples were loaded on the same 10X chip to minimize potential batch effects. Single-nucleus libraries from individual samples were pooled and sequenced on the NovaSeq 6000 instrument (Illumina) with an average depth of ~10,870 unique molecular identifiers (UMIs) per nucleus for snRNA-seq and ~8223 unique fragments per nucleus for snATAC-seq.

## Fluorescence-activated nuclei sorting (FANS) and nuclear RNA-seq of the three populations of the human brain cells (neurons, oligodendrocyte lineage cells, and other glia)

The isolation of brain nuclei prior to the flow cytometry separation was based on the previously published protocol[88]. Tissue (~300 mg) was homogenized in ice-cold lysis buffer (320 mM sucrose, 5 mM CaCl$_2$, 3 mM Mg(Ac)$_2$, 0.1 mM EDTA, 0.1% Triton X-100, 1 mM DTT, 1 U/μl RRI, and 10 mM Tris-HCl, pH 8.0), underlaid with the sucrose buffer (1.8 M Sucrose, 3 mM Mg(Ac)$_2$, 1 mM DTT, 0.4 U/μl RRI, and 10 mM Tris-HCl pH 8.0), and centrifuged for 1 h at 98,600 g (24,000 rpm using SW41Ti rotor). The nuclear pellets were then resuspended in the antibody-incubation buffer (0.5% BSA, 3 mM MgCl$_2$, 1 U/μl RRI, and 10 mM Tris-HCl, pH 8.0) and incubated with antibodies against NeuN and SOX10 for 1 h at 4 °C. FANS method was then used to isolate neurons (NeuN+SOX10- population), oligodendrocyte lineage cells (NeuN-SOX10+ population), consisting of mature oligodendrocytes and a smaller population of OPCs, and a third population (NeuN-SOX10-; hereafter "other glia") that mostly consisted of astrocytes and microglia.

NeuN (also known as RNA-Binding Protein RBFOX3) is a well-established marker of neuronal nuclei[89]. SOX10 is a transcription factor specifically expressed in oligodendrocyte lineage cells. The application of anti-SOX10 antibodies to isolate oligodendrocyte lineage nuclei was described by Frisen and colleagues[90]. We used Alexa488-conjugated anti-NeuN antibodies (1:1000 dilution, Millipore, Cat. # MAB377x) and anti-SOX10 antibodies (R&D Systems, Cat. # AF2864) that were custom conjugated to Alexa647 (1:150 dilution, Cat# FCMAB317PE, Millipore). DNA stain DAPI was used to label intact nuclei (see Supplementary Fig. 5 for details). From each sample, we collected 250-300 thousand (K) nuclei of neurons, 250–300 K of oligodendrocyte lineage cells, and 150-200 K nuclei of other glia. Nuclei were collected directly to the lysis buffer from PicoPure RNA Isolation Kit (ThermoFisher Scientific, Cat. # KIT024), which was then used for RNA isolation. RNA-seq libraries were prepared with the SMARTer Stranded Total RNA-seq Kit, Pico-Input v2 (Takara, Cat. # 634414) from 10 ng of the RNA. Libraries were sequenced on a NovaSeq 6000 instrument (Illumina), using paired-end 100 cycles protocol to an average of 120 million read pairs per sample.

## FANS and H3K27ac ChIP-seq of the four populations of the brain cells (neurons, oligodendrocyte lineage, astrocytes, and microglia)

FANS protocol to isolate neurons, oligodendrocyte lineage, astrocytes, and microglia was performed as described in the previous section, except removing RRI from all buffers, adding 0.1 mM benzamidine, 0.1 mM phenylmethylsulfonyl fluoride (PMSF) to the lysis buffer, and adding antibodies against interferon response factor 5 (IRF5). IRF5 is highly enriched in cells of myeloid origin, including microglia[91]. The IRF5 antibodies allowed the separation of astrocytes and microglia within the other glia (NeuN-OLIG10-) population (see Supplementary Fig. 5 for details). In this FANS protocol, we used PE-conjugated anti-NeuN antibodies (1:1000 dilution, Millipore, Cat. # FCMAB317PE), anti-SOX10 antibodies (R&D Systems, Cat. # AF2864), which were custom-conjugated to Alexa647 (1:150 dilution), and Alexa488-conjugated anti-IRF5 antibodies (1:200 dilution, R&D Systems, Cat. # IC4508G).

We employed the native ChIP protocol (N-ChIP) in which chromatin fragmentation is performed using micrococcal nuclease

(MNase) without crosslinking proteins to DNA[92]. 100-150 K of each cell type were collected and used for each ChIP reaction. We used anti-H3K27ac antibodies from Active Motif (Cat# 39133; rabbit polyclonal, 3 μg per sample). ChIP-Seq libraries were prepared with the NEBNext Ultra II DNA Library Prep Kit for Illumina (New England Biolabs, Cat. # E7645). The resulting libraries were sequenced on a NovaSeq 6000 instrument (Illumina), using paired-end 100 cycles protocol, to an average of 60 million read pairs per sample. For each diagnostic group, three input control samples obtained from MNase-digested DNA were prepared and sequenced.

## Jess/Wes automated, multiplex Western blot assay

Capillary Western analyses were performed using the ProteinSimple Jess-Wes System (San Jose, CA, USA). The following reagents were used: Jess/Wes-Separation Module [2-40 kDa Kit (Cat. # SM-W012) 12-230 kDa Kit (Cat. # SM-W004) and 66-440 kDa kit (Cat. # SM-W007)]; Jess/Wes- anti-rabbit, anti-mouse, and anti-goat Detection Module kit (Cat. # DM-001, DM-002 and DM-006, respectively); anti-mouse, anti-rabbit and anti-goat secondary antibodies (Cat. # 042-205, 042-206 and 043-522-2, respectively); antibody diluent (Cat # 042-203); EZ standard Pack 1, 3 and 5, containing biotinylated ladder (molecular weight 12-230 kDa, 66-440 kDa, and 2-40 kDa, respectively); fluorescent (FL) standards, containing 29 kDa (Cat. # PS-ST01EZ), 90 kDa (PS-ST03EZ), and 26 kDa (PS-ST05EZ) system controls; dithiothreitol (DTT); streptavidin-HRP; luminol-S, peroxide; sample buffer (Cat. # 042-195); and wash buffer (Cat. # 042-202). The Separation Module kit included capillary cartridge and pre-filled microplates. Primary Antibodies used were specific for HSP90 (Mouse, R&D Systems Cat, # MAB3286; Clone # 341320), GFAP (Mouse, Sigma Cat, # G3893), CD44 (Mouse, R&D Systems, Cat. # BBA10; Clone # 2C5), CHI3L1 (Goat, R&D Systems, Cat. # AF2599), HSP70 (Rabbit, R&D Systems, Cat. # AF1663), HSP27 (Rabbit, R&D Systems, Cat # AF1580), RANBP3L (Rabbit, Novus Cat # NBP2-38347), KCND3/Kv4.3 (Rabbit, Novus Cat. # NBP2-76945), DNMT3A (Mouse, Novus Cat # 64B1446), NEFL (Mouse, R&D Cat # MAB22163; Clone #1002615), Clusterin (Mouse, Novus Cat. # MAB2937, clone 3500227), RAP1GAP (Rabbit, Novus Cat. # NBP1-53072), UbiquitinB (Mouse, Novus Cat. # NBP3-07163, clone UBB/1748), TGFB2 (Goat, Novus Cat # AB-112-NA), MAOB (Rabbit, Novus Cat # NBP3-15411, clone 5M8A5), ATP5A (Rabbit, Novus Cat # NBP2-92928), Cytochrome C (CYCS) (Rabbit, Novus Cat # NBP2-21569), and COX5A (Rabbit, Novus Cat # NBP1-32550).

The assay was performed in the motor cortex samples from C9-ALS and control donors (Supplementary Dataset 1). For each subject, ~80–100 mg of tissue were homogenized in lysis buffer (3 μl/mg of tissue) [0.32 M Sucrose, 5 mM CaCl2, 3 mM Mg(Ac)2, 150 mM NaCl, 0.1 mM EDTA, 0.1% Triton X-100, 3% Igepal.NP-40 (v/v), 1% sodium deoxycholate (w/v), 0.1% SDS (w/v), 1 mM PMSF, Protease Inhibitor Cocktail (Sigma, St. Louis, MO; 1:100 v/v), and 10 mM Tris (pH 8.0)] using a ME220 focused ultrasonicator (Covaris, Cat # 500506). Burst setting of 20 s, power setting of 180 watts were used; the suspension was chilled at 4 °C between ultrasonic bursts. Following homogenization, an additional 0.2 mM of PMSF was added to each sample, and the samples were incubated on ice for 30 min. The homogenates were then centrifuged at 10,000 g for 20 min at 4 °C, and the supernatant (whole-tissue extract) was transferred to a new tube. Total protein concentrations of the whole-tissue extracts were measured using Qubit Protein Assay (ThermoFisher Scientific, Cat # Q33211). Samples were stored at −80 °C.

On the day of an experiment, samples (whole-tissue extracts) were defrosted and diluted with 0.1X Sample Buffer (from a 10X stock). The diluted samples were then combined with 5X Fluorescent (5X FL) Master Mix (containing 5X sample buffer, 5X fluorescent standard, and 200 mM DTT) in a 4:1 ratio and denatured by heating at 95 °C for 5 min using TruTemp DNA Microheating System (Robbins Scientific, Cat # 1057-30-0). The Fluorescent Master Mix contains three fluorescent

proteins that serve to normalize the separation distance within each capillary, as the molecular weight ladder is loaded only to the first capillary. Following the denaturation step, the samples, blocking reagent, chemiluminescent substrate (mixture of luminol-S and peroxide in a 1:1 ratio), washing buffer, HRP-conjugated secondary antibodies, and a primary antibody were dispensed into designated wells in the assay plate. Following plate-loading, the samples were subjected to a fully automated capillary separation electrophoresis and immunodetection. One target protein was tested in each assay, and the corresponding primary antibodies were diluted to ensure the linear dynamic range of detection for each protein. The following dilutions were used: 1:5 dilutions for clusterin (CLU), 1:10 dilution for CHI3L1, CD44, HSP90, KCND3/Kv4.3, DNMT3A, RAP1GAP, Ubiquitin B, COX5A and 1:20 dilution for GFAP, HSP27, NRFL, RANBP3L, MAOB, ATP5A, CYCS.

For the statistical analysis, ProteinSimple Compass for SW software was used to extract areas under peak values, which were used to analyze the immunoreactive signals. A two-sided Welch's $t$-test was used to compare the signals between C9-ALS and control, where the averaged signals between two replicates for each donor were used as observations (6 donors each for C9-ALS and control), and the comparisons with a $p$-value <0.05 were deemed significant.

## GFAP immunofluorescence

Frozen human postmortem motor cortex blocks from control and C9-ALS samples were embedded in OCT (TissueTek Sakura) and cryosectioned at 10 μm at −20 °C (Thermo Cryostar). Sections were placed onto Superfrost Plus glass slides (Fisher Scientific), dried for 20 min at −20 °C, sealed, and stored at −80 °C until use. Immunofluorescence staining was performed in accordance with the previously published protocol[93]. Sections were washed in 1x PBS for 60 min, treated with 0.5% sodium borohydride to remove aldehydes, rinsed in PBS, incubated for 10 min in 3% hydrogen peroxide to inhibit endogenous peroxidase, and then incubated in blocking solution (1xPBS, 0.3% Triton X-100, 10% Normal Goat Serum, 1% Bovine Serum Albumin) for 2 h at room temperature (RT). This was followed by overnight incubation at 4 °C in blocking solution with rabbit anti-GFAP primary antibodies (1:1000; Millipore Sigma, Cat. #AB5804). Sections were then washed 3 times in 1x PBS (10 min each) and incubated with goat anti-rabbit IgG Alexa Fluor 594 secondary antibodies (1:500, Invitrogen). Subsequently, sections were incubated in 0.1% Sudan Black in 70% ethanol for 5 min at RT to suppress autofluorescence, washed in Tris buffer saline (pH 7.6) 3 times (15 min each), cover-slipped with ProLong Gold antifade reagent with DAPI (Invitrogen), and seal with nail polish. Fluorescent labeled sections were imaged using 63x oil-immersion objectives. The images were captured on an inverted Zeiss LSM 700 confocal microscope (Zeiss, Germany). Double labeling was confirmed by acquiring z-stacks at 1 to 0.5 μm intervals through cells of interest and by maximum intensity projection (MIP) views. The images were acquired in 2 control and 2 C9-ALS subjects, 3 sections per subject and 3 fields in each section.

## snRNA-seq data preprocessing, clustering and cell type annotation

snRNA-seq reads were mapped to the human GRCh38 genome with a custom pre-mRNA annotation (modified RefSef gene annotation where each gene transcript locus was listed as an exon) using 10x Genomics Cell Ranger (v3.0.2)[94] with the default parameters. The raw feature-by-droplet matrices generated from the "cellranger count" module were then fed to CellBender (v0.2.0)[95] to distinguish cell-containing from cell-free droplets and retrieve noise-free cell-by-gene quantification tables using the following parameters: "--expected-cells 1500 --total-droplets-included 18,000 --model full --epochs 150 --cuda --low-count-threshold 15 --fpr 0.01 --learning-rate 0.0001 --posterior-batch-size 5 --cells-posterior-reg-calc 50".

Scrublet (v0.2.3)[96] was then used in the CellBender-generated h5 file for each sample to compute the doublet score for each nucleus with an expected doublet rate of 0.06. Based on the simulated doublet histogram of doublet scores, we set the doublet score cutoff to 0.2 and this threshold worked well for all samples.

Next, Seurat (v4.0.4)[97] was used to create a merged object containing all the samples, and to perform the following downstream processing. Nuclei with <500 detected genes, >1% reads that mapped to the mitochondrial genome, or a doublet score >0.2 were excluded from further analysis. Sctransform[98] normalization was performed on the remaining nuclei, with the mitochondrial mapping percentage and the sequencing batches set as the confounding sources of variation to be removed. Principal component analysis (PCA) was run with the top 3000 highly variable genes in the normalized nuclei-by-gene matrix, and the first 30 principal components were used to run Harmony (v0.1.0)[99] integration with sample IDs, donor IDs and sequencing batches set as covariants. Then, a k-nearest-neighbor graph was constructed with k set as 30, and the nuclei were clustered using Leiden clustering with a resolution of 1. This gave rise to 29 clusters, and UMAP (Uniform Manifold Approximation and Projection)[100] was used for visualization (Fig. 1a, the leftmost panel).

The nuclei were then classified into three classes (excitatory neurons, inhibitory neurons, and non-neurons) using several well-established marker genes (Supplementary Fig. 3a). For each of these three classes, a subset of nuclei was created from the full dataset, and the aforementioned Sctransform normalization, PCA, Harmony integration and Leiden clustering steps were run on each of these subsets. This iterative clustering process reselects the highly variable genes in the context of each cell class and hence may give better resolution in finding distinct subclusters. We used SC3 stability index[101] and Clustree (v0.5.0)[102] to evaluate and visualize the stability of the clusters across a range of resolutions for the Leiden community detection analysis (Supplementary Fig. 2f). With the Leiden resolution set as 1.5, 0.5 and 1, we identified 24, 23, and 23 subclusters for excitatory neurons, inhibitory neurons and non-neurons, respectively. These subclusters were annotated using the expression of selected marked genes (Supplementary Fig. 3b–d). Note that six subclusters seemed to express marker genes of multiple cell classes (for example an excitatory subcluster with expressions of oligodendrocyte marker genes), which were potential doublets that failed to be captured by the Scrublet pipeline. We labeled these subclusters as "ambiguous" and removed them from further consideration. We also noticed one excitatory neuron subcluster that highly expressed the *TUBB2A* gene but had a low number of detected genes, and one non-neuron subcluster that expressed marker genes of immune cells (but no expression of microglia marker genes). We also excluded these two subclusters from the downstream analysis. After annotation, we identified 49 distinct subpopulations of excitatory neurons, inhibitory neurons and non-neuronal cells. To estimate how well our annotation matched with the cell types annotated in the recent human motor cortex single nuclei study by the Allen Institute for Brain Science[24], we used Seurat's "FindTransferAnchors" and "TransferData" functions to transfer the annotation labels from the AIBS reference dataset to our dataset. A confusion matrix was constructed with the prediction labels (label with the highest prediction score) from the data transfer and the labels from our annotation, and the adjusted rand index was calculated to measure the similarity between the two (Supplementary Fig. 4). Cell type nomenclature mostly followed the common cell type nomenclature (CCN) convention presented by the Allen Institute for Brain Science[24,103], with slight modifications to reflect the specific marker genes found in our dataset.

### Differential expression analysis in snRNA-seq
We first performed unsupervised analysis to characterize the similarity of samples from each patient in an unbiased manner.

We created pseudobulk counts with nuclei originating from each donor for each cell type, computed normalized counts with DEseq2's median of ratios method (v1.36.0)[104], and performed PCA separately for each cell type and each brain region using the log normalized counts (Supplementary Fig. 14). Although the number of subjects is limited, we did observe a separation of the disease and control samples, especially in astrocytes and excitatory neurons. The inter-individual variability of gene expression was greater in some cell types than others. Second, we used Augur (v1.0.3)[28] to quantify the separability of nuclei from different diagnoses. Augur employs a machine-learning framework to quantify the separability within a high-dimensional space. For each cell type, Augur withholds a proportion of sample labels and trains a random forest classifier on the labeled subset. The classifier predictions are compared with the experimental labels, and cell types are prioritized on the basis of the area under the receiver operating characteristic (AUC) of these predictions in cross-validation. This analysis showed that we can distinguish nuclei from disease or control samples using the gene expression profile, with the highest AUC of ~0.78 achieved in astrocytes for C9-ALS vs. control (Supplementary Fig. 15a). Furthermore, for the C9-ALS vs. control analysis, we generated random shuffles to check how much better these observed AUCs would perform than the shuffles. We permuted the diagnosis labels of the individual donors (so that all nuclei from each donor will still have the same diagnosis labels), and required that the two shuffled groups must each contain three C9-ALS donors and three Control donors, of which three are male and three are female. This resulted in 82 possible permutations, and we ran Augur to get the AUCs for distinguishing the two groups. We found that in motor cortex the AUCs from the real diagnosis labels are higher than the 95% upper confidence intervals in astrocytes, excitatory neurons and VIP inhibitory neurons. Astrocytes and deep-layer excitatory neurons also have higher AUCs in the observed than those in the shuffles in the frontal cortex samples (Supplementary Fig. 15b). Last, we applied linear discriminant analysis (Supplementary Fig. 15c) to quantify the discriminability of C9-ALS cells vs. control cells. We separated the data into training and test sets based on the donor, using 10 donors for training and 2 for testing (one each from C9-ALS and control). We trained a LDA classifier using the top 1-10 principal components for each cell type and brain region. The results showed that C9-ALS cells could be meaningfully distinguished from control using the top few principal components in deep and superficial excitatory neurons, and in astrocytes.

Next, we performed the traditional differential expression analysis with the labels of diagnosis. To ensure a statistically meaningful number of cells from each donor for the differential expression analysis, we grouped the subpopulations into 14 major cell types with shared marker genes. In these major cell types, there were at least 15 high-quality nuclei in at least 4 out of the 6 donors in each group (the combination of diagnosis and brain region). For the comparison between C9-FTD and control samples, we focused on the non-neuronal cell types as the C9-FTD samples showed depletion in the numbers of neuronal nuclei (Fig. 5a). To identify the differentially expressed genes (DEGs) between disease (C9-ALS or C9-FTD) and control across these major cell types in each brain region, we used the model-based analysis of single-cell transcriptomes (MAST, v1.18.0)[29], where a mixed-effect hurdle model was employed to model the snRNA expression data as a mixture of a binomial and normal distribution while systematically accounting for predefined covariates. For each comparison, the raw counts of chosen nuclei in question were extracted from the "RNA" slot of the Seurat R object and then normalized to $\log_2$CPM (counts per million). Genes that were expressed in at least 10% of the selected nuclei were kept. The following linear mixed model was then

fit with MAST:

$$y \sim D + G + U + M + A + S + B + (1|I) \tag{1}$$

Here, $y$ is the $\log_2$-normalized count of the gene; $D$ is the diagnosis (ALS, FTD or control); $G$ is the number of genes detected, $U$ is the number of UMIs; $M$ is the percentage of reads mapped to mitochondrial genes; $A$ is the age of the donor; $S$ is the sex of the donor; $B$ is the sequencing batch of the sample; and $I$ is the ID of the donor. $G$, $U$, $M$ and $A$ were centered and scaled across the selected nuclei in comparison. The donor ID $I$ was modeled as a random-effect term while the rest of the covariates were modeled as fixed-effect terms.

Next, a likelihood ratio test (LRT) was performed to identify DEGs by comparing the model with and without the diagnosis term. Hurdle $p$-values were reported by MAST, and Benjamini & Hochberg's false discovery rate (FDR) method was used to adjust $p$-values for multiple comparisons. The MAST model also reported fold-changes (FC) due to the disease effect (hereafter referred to as model FC), which were contributed by both the continuous component (nonzero expression) and the discrete component (expressed or not) of the hurdle model. We also computed more straightforward fold changes between the groups (hereafter referred to as average FC) by subtracting the mean $\log_2$CPM of selected nuclei in the control sample from the mean $\log_2$CPM of nuclei in the disease sample. In most cases these two fold-changes were concordant, and we excluded genes with extremely high model FC that deviated from the average FC. Genes were defined as significantly differentially expressed if they passed all of the following criteria: the FDR was <0.05; the absolute model FC was >20%; both the continuous and the discrete fits of the hurdle model were convergent; the upper and lower bounds of the 95% confidence model had the same sign; the difference between the $\log_2$(model FC) and $\log_2$(average FC) was <2.

To control for the number of nuclei when comparing the disease effect across the major cell types, we randomly downsampled the dataset to 30 nuclei per donor in each cell type, and performed the same MAST test as described above. The downsample analysis was done 10 times, and the number of significant DEGs for each cell type was reported as mean ± SEM (standard error of the mean).

To avoid double dipping when comparing the disease effect in non-neuronal cells between C9-ALS and C9-FTD, we split the 6 control donors into two groups (group 1: C1, C2 and C3, two males and one female; group 2: C4, C5 and C6, one male and two females). The same MAST test was run as described above, except that the group 1 control donors were used in the C9-ALS vs. control comparison while group 2 control donors were used in the C9-FTD vs. control comparison. The overlaps of significant DEGs between the two comparisons were reported, and the correlations of model fold-changes were estimated using the Pearson correlation coefficient.

## Gene Ontology (GO) enrichment test

GO enrichment analysis was done using WebGestalt (v2019)[36]. Up- and Downregulated DEGs identified in each cell type and each comparison were used separately as input, and all the expressed genes in the corresponding group were set as the reference gene set (background). GO terms in the functional databases "Biological Process noRedundant", "Cellular Component noRedundant" and "Molecular Function noRedundant" were used, and only terms with 25 to 500 genes were included in the over-representation analysis. Significantly enriched GO terms (FDR < 0.05) were then clustered by Affinity Propagation to further remove reductant terms.

To compare the effect size of differential expression between motor cortex and frontal cortex, we calculated the fold-change difference (Δ $\log_2$FC) between the two brain regions for genes associated with each GO term of interest. The Δ $\log_2$FC for all expressed genes (labeled as "all genes" in Fig. 4d and Supplementary Fig. 8a) and all DE genes (labeled as "DE genes") were also computed and used as baseline control. The two-sided Welch's $t$-test was used to test whether the Δ $\log_2$FC in each group of genes were significantly different from the Δ $\log_2$FC of the "all genes" control set.

## Bulk FANS-sorted nuclear RNA-seq data processing

FastQC (v0.11.8; https://www.bioinformatics.babraham.ac.uk/projects/fastqc/) was used to examine the quality of the RNA-seq reads. Reads were then trimmed to remove sequencing adapters and low-quality sequences (minimum Phred score 20) using Trim Galore (v0.5.0, a wrapper tool powered by Cutadapt[105]) in paired-end mode. The first 3 bp from the 5′ end of read 1 were also removed. Trimmed reads were then mapped to the human hg38 genome and the GEN-CODE annotated transcriptome (release V35) with STAR (Spliced Transcripts Alignment to a Reference, v 2.7.1a)[106]. Gene expression was estimated using RSEM (RNA-Seq by Expectation Maximization, v1.2.30)[107]. Gene-level 'expected count' from the RSEM results were rounded and fed into edgeR (v3.34.1)[108] to perform differential expression tests separately for each sorted population in each brain region. Only genes that were expressed (with CPM > 2) in at least six samples were kept. The read counts of the remaining genes were then normalized using the TMM method[109], and DE genes were called in the quasi-likelihood F-test mode with a cutoff of FDR < 0.05 (Supplementary Dataset 13). No DE genes were detected at this FDR cutoff, and we only used the fold-changes in the bulk RNA-seq data to perform the correlation test with the fold-changes in snRNA-seq.

## snATAC-seq data preprocessing, clustering and cell type annotation

snATAC-seq reads were mapped to the human GRCh38 genome using 10x Genomics Cell Ranger ATAC (v1.1.0)[110] with the default parameters. Barcode multiplets[111] were removed using the "clean_barcode_multiplets_1.1.py" script provided by 10x Genomics. The output fragments files were then imported into ArchR (v1.0.2)[112] to create Arrow files and an "ArchRProject" object for the downstream analysis. Fragment size distribution in each sample was inspected for nucleosomal periodicity, and nuclei with TSS (Transcription Start Site) enrichment score <4 and a number of unique fragments <1000 were removed. Doublet inference and removal were performed using the "addDoubletScores" and "filterDoublets" functions in ArchR. A cell-by-tile matrix containing insertion counts across genome-wide 500-bp bins was created, and dimensionality reduction was applied to the matrix using the iterative Latent Semantic Indexing (LSI) implemented in ArchR (two iterations with the top 25,000 features). The top 30 dimensions after LSI were used to perform Louvain clustering (with resolution set as 2) and UMAP visualization.

Next, a cell-by-gene matrix was computed with the gene activity score model implemented by ArchR. The model accounted for both the accessibility within the entire gene body and the activity of putative distal regulatory elements using an exponential weighting function that depends on the distance between the insertions and the TSS of the gene of interest, and gene boundaries were imposed to minimize the contribution of unrelated regulatory elements to the gene activity score. Gene activity scores of well-defined cell class marker genes were used to classify the clusters into excitatory neurons, inhibitory neurons and non-neurons. A subset of nuclei was created for each of these three classes, and a similar LSI and clustering pipeline was run to identify subclusters in each category. For visualization, the "addImputeWeights" function was used to impute gene activity scores by smoothing signals across nearby cells based on a MAGIC[113] diffusion matrix.

To annotate these subclusters, we integrated the gene activity score matrix in the snATAC dataset with the gene expression matrix in our snRNA dataset using the "addGeneIntegrationMatrix" function. The integration was done in a constrained way, where subclusters from

the three cell classes in the snATAC dataset were only allowed to align to the subclusters with matching cell classes in the snRNA dataset. The annotation labels from the 14 major snRNA cell types were transferred to each snATAC nuclei with a prediction score to represent the assignment accuracy. The final annotations of the snATAC subclusters were determined by first filtering out nuclei with a prediction score <0.7 and then taking the transferred label of 70% supermajority in remaining nuclei for each cluster. Subclusters that failed to reach the supermajority were labeled as "Mixed" and removed from further consideration. This resulted in 109,198 high-quality snATAC nuclei of 11 major brain cell types.

## snATAC-seq peak calling

To call peaks in each group of interest (the combination of major cell types, brain regions and diagnosis), we first created pseudo-bulk replicates using the "addGroupCoverages" function in ArchR with the following settings: the minimum and maximum numbers of replicates were set to 6 and 28; the minimum and maximum numbers of nuclei per replicate were set to 50 and 500; the sampling ratio to use if a particular group lacks sufficient cells to make the desired replicates was set to 0.8. Next, the "addReproduciblePeakSet" in ArchR was run with MACS2 (v2.2.7.1)[114] set as the peak calling method, where the iterative overlap peak merging procedure was performed to generate a single merged peak set of fixed-width (501 bp), reproducible peaks that can be called in at least two samples. A cell-by-peak count matrix was then computed with the "addPeakMatrix" function. The counts were normalized by the number of reads in TSS across cells prior to performing marker feature identification, and the cell-type-specific peaks were identified using the Wilcoxon test in the "getMarkerFeatures" function for each major cell type, adjusting for the potential bias introduced by the number of unique fragments and TSS enrichment.

## Differential chromatin accessibility between disease and control

Differential chromatin accessibility between disease and control was accessed in two ways. First, we ran gene-based pair-wise differential tests with the gene activity scores for each major cell type using the Wilcoxon test in the "getMarkerFeatures" function, accounting for bias of the number of unique fragments and TSS enrichment. Genes with FDR <0.05 were called significant. In addition, peak-based differential tests were performed to identify differentially accessible regions (DARs) between disease and control for each major cell type. The cell-by-peak matrix was normalized by the number of reads in TSS across cells and DARs were identified using the Wilcoxon test in the "getMarkerFeatures" function, adjusting for the potential bias introduced by the number of unique fragments and TSS enrichment. No significant DARs were found with an FDR threshold of 0.05.

## Accessibility of transcription factor motif

ChromVAR[67] was used to estimate chromatin accessibility within peaks sharing the same transcription factor (TF) motif while controlling for technical biases. First, the non-redundant transcription factor (TF) motif archetypes (v2.0-beta, https://github.com/jvierstra/motif-clustering)[115] were used to scan and annotate the peaks with the "addMotifAnnotations" function in ArchR. Background peaks were selected based on similarity in GC content and the number of fragments across all samples using the "addBgdPeaks" function. The chromVAR deviations and z scores per cell were then computed for each TF motif using the "addDeviationsMatrix" function. To find TF motifs with differential chromVAR scores between disease and control in each major cell type, t-test and Brown-Forsythe test were performed on the deviations and z scores, respectively. Motifs with FDR < 0.05 were called significant. Motif sequence logos were drawn using the "ceqlogo" module in the MEME Suite (v5.4.1)[116].

## H3K27ac ChIP-seq data processing

Raw sequencing data were pre-processed to remove adapters and low-quality sequences with the HTStream tool (https://github.com/s4hts/HTStream). Reads were mapped to the human hg38 genome build with BWA-MEM2[117] and filtered to remove multi-mapping reads and low-quality alignments using Samtools[118]. Reads mapping to ENCODE-blacklisted genomic regions were excluded using BEDTools[119]. H3K27ac-enriched peaks were detected using MACS2, including input controls for each cell type and condition, as previously described[120]. Promoter (2 kb upstream and 1 kb downstream from TSS) H3K27ac signal was computed using the "multiBigwigSummary" module from deepTools (v.3.3.1)[121]. Differential peaks between C9-ALS and control for each FANS-sort population were identified using DiffBind[122] in edgeR[108] mode. Peaks with FDR < 0.05 were called significant.

## Reporting summary

Further information on research design is available in the Nature Portfolio Reporting Summary linked to this article.

# Data availability

All sequencing data generated in this study have been deposited in the Gene Expression Omnibus repository under the accession code GSE219281. Curated snRNA-seq cell-by-gene tables are provided in the following Zenodo repository:[123] https://doi.org/10.5281/zenodo.8190317. Metadata for each nucleus in snRNA-seq and snATAC-seq are provided in Supplementary Dataset 2 and Supplementary Dataset 6. An IGV browser session showing ChIP-Seq, snRNA-seq, and snATAC-seq tracks, and a UCSC single cell browser session showing the snRNA-seq data, are available at https://brainome.ucsd.edu/C9_ALS_FTD/. Source data are provided with this paper.

# Code availability

The codes used in this study are available at the following Github repository linked to Zenodo: https://github.com/hoholee/C9_ALS_FTD_single_nuclei_transcriptome_epigenome (https://doi.org/10.5281/zenodo.8188162)[124].

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

## Acknowledgements

We gratefully acknowledge Eugene Koonin for helpful comments, and Ramu Vadukapuram for performing nuclear sorting for ChIP-seq studies. This work was supported by VA Merit Awards BX003625 and BX005585 (S.D.); PsychEncode consortium NIH/National Institute of Mental Health R01MH122590 (former U01MH122590) (S.D.) and R01MH122592 (former U01MH122592) (E.A.M.); R01AG067151 (V.V.B.). We gratefully acknowledge Mayo Clinic for its financial support of "The Mayo Clinic Brain Bank". We also gratefully acknowledge Drs. Boylan and Oskarsson, who are clinicians who were essential in obtaining C9-ALS autopsies and the Einstein Aging Study (P01AG003949) for providing tissue of 3 of the 11 control samples in our study.

## Author contributions

The study was conceived and designed by J.L., V.V.B., E.A.M. and S.D. J.L. performed computational analyses of all sequencing data, and generated the majority of figures and tables. M.K.J. performed snRNA-seq, snATAC-seq, FANS-sorted bulk RNA-seq, ChIP-seq, and GFAP immunofluorescence experiments, and generated the GFAP immunofluorescence figure. J.C. participated in computational analysis, performed GO analyses and generated GO-related figures. A.K. designed and performed FANS-sorted ChIP-seq experiments, and processed the ChIP-seq data. M.K.J. and J.J. performed and analyzed digital Western Blot validation experiments. P.Z. assisted in single cell and FANS-sorted sequencing experiments. M.G. and L.J.P. performed dissection of the motor and prefrontal cortices. D.W.D. and E.E.-C. performed cases chart review to provide clinical information. V.V.B. and D.W.D. provided biospecimens. J.L., J.C. and E.A.M. designed and built the data browser. J.L., S.D., and E.A.M. interpreted the results and wrote the original draft of the manuscript. S.D. and E.A.M. supervised and provided funding for the study. All authors read, edited and approved the final manuscript.

## Competing interests

The authors declare no competing interests.
