## [Peer Review File · Nature Communications]

REVIEWER COMMENTS

Reviewer #1 (Remarks to the Author):

Li et al provide a single cell omic analysis of C9orf72 ALS/FTD and healthy controls, including transcriptomics, ATACseq and H3K27Ac CHIP. snRNAseq from ALS is by far the most robust dataset, because the C9-FTD samples appear to have insufficient quality for a full analysis (neuron data is missing). Some hits are validated by immunoblotting. Comparing frontal vs motor cortex makes a lot of sense, although the results are surprisingly similar. The epigenetic methods are ambitious but provide limited additional information, because they seem statistically underpowered. Nevertheless, the manuscripts will be a valuable resource for the community and should be published after revision.

Major points:

-From the abstract it is unclear to what the C9orf72-ALS and FTD cases are compared. To each other, to non-C9-ALS/FTD or to healthy controls? This is answered early in the paper, but the title and abstract insinuate C9-specific effects are analyzed, which is not the case, because no ALS/FTD cases without the mutation were analyzed. The nature of the controls becomes clear early in the text but the limitations are not mentioned anyway, e.g. effects on translation are directly attributed to DPRs, although the same changes may occur in non-C9-ALS/FTD. Probably the majority of the identified changes are also present in sporadic ALS/FTD, which has very similar clinical presentation. See competing preprint (<https://www.biorxiv.org/content/10.1101/2021.07.07.451374v2.full>). While it makes sense to focus on C9orf72 because they are potentially more homogenous than sporadic ALS and FTD, but without comparison to sporadic ALS/FTD the interpretation will also be limited. Without such new data the title should be revised.

-There is significant overlap with a preprint that actually compared C9orf72 vs sporadic ALS/FTD in a larger cohort (<https://www.biorxiv.org/content/10.1101/2021.07.07.451374v2.full>), which should be discussed and analyzed based on the data available on biorxiv. The present manuscript uses far less sample but also looks at epigenetics.

-Fig 2G: Since the control and C9-ALS samples differ by genotype and disease status it is impossible to tell whether the C9orf72 repeat expansion triggers a cell type specific reduction in C9orf72 expression or whether a general suppression is overlaid by ALS-specific upregulation, e.g. in microglia.

-Validate GOLGB1 and especially MAOB by western blot, because large part of the discussion is based on it.

-Validate mitochondrial hits by western blot given conflicting results showing reduction of mitochondria in ALS patients using EM.

-Fig 6B: Is neuron loss really that much more severe in FTD than in ALS? Quantify neuron loss in samples by other means, e.g. IHC or flow cytometry.

-Elaborate on differences ALS and FTD and discuss selected genes

Minor points:

-What do the box plot next to the GO dotplots mean? e.g. Fig. S8A

-Even though the clustering seems thorough and valid, it is unclear how the authors determined the clustering resolution. Usage of clustree is highly recommended.

-Please provide specific version of Seurat that were used for this analysis. More information is promised in supplementary methods, but I could not find it.

Reviewer #2 (Remarks to the Author):

The authors describe an extensive single-nucleus-based transcriptomic and epigenomic characterization of post-mortem motor and frontal cortices in C9-ALS and C9-FTD cases. The data are of high quality and the analyses are competent. There are some intriguing findings that suggest the different types of neural cells may be differentially affected in different regions of the brain, and that this may depend on the clinical and pathological presentation of the disease. The authors go on to discuss the implications of these findings. Overall, this would be a valuable resource to the community, although the studies could be strengthened further.

Key concerns include the following:

1. In the absence of comparisons with other neurodegenerative diseases, it is difficult to see how many of these findings are specific to ALS-FTD and how many are general responses to a degenerative process (e.g., changes in astrocytes – are the changes described specific to this disease?).

2. Were the C9-ALS cases specifically assessed for cognitive dysfunction and was it confirmed to be absent? Were these cases also assessed post-mortem to confirm absence of pathology in non-motor regions? It would be helpful to clarify this in the manuscript (apologies if the authors have done so and I missed it).

3. If possible, it would be helpful to test some of the speculations in the discussion on post-mortem tissue – e.g., would you expect to see an upregulation of translation system components close to where there is evidence of DPRs, or an upregulation of ROS scavengers in neurons with signs of DNA damage? Could this be tested by IF experiments? If yes, is it generalizable or specific to ALS-FTD?

Minor points:

1. The transcriptomic and epigenomic data are analyzed separately – would it be possible to assess them together, to potentially highlight transcriptional regulatory circuits?
2. There is a lot of descriptive data here, which I found difficult to keep track of – it would be helpful if there were a schematic/graphical/cartoon representation of the various cell- and layer-specific changes in the different cortical regions, and their relationship to the clinical and pathological presentation of the disease. It would help the readers to better absorb and appreciate the data.

Reviewer #3 (Remarks to the Author):

This is a thorough analysis of the frontal and motor cortex in patients with C9ORF72 mutations displaying either FTD (n=5) or ALS (n=6) compared to control cases (n=6). The patient groups were generally well matched for age, sex and race. This is certainly a fantastic resource paper but not a primary research paper as there is no validation of any of their results or mechanistic insight into the snSEQ analysis. This also leads to another problem which is that many of the results do not support their conclusions without lack and gain of function experiments. In fact the authors say in the caveats section “we cannot distinguish cell-type-specific alterations that represent direct effects of C9 mutation from end-of-life consequences of neurodegeneration”. And yet in the paper they do make assumptions from this data set. This was the major weakness of the paper. But the data have been generated in a very reliable way and would be of great value to the community.

1. The major weakness of this paper is that the authors move directly to supervised statistical methods to look for DEGS between C9 and control in specific genes/cells rather than a pseudobulk PCA analysis to see if ALS and control reliably separate without prior labeling. The history of this type of analysis is that there will always be DEGS in a single data set that are significant when looking at this many groups of cells and numbers of genes. While FDR was used this is prone to issues across this many cell types and genes. The gold standard would be to run a validation set of samples to see if the same gene sets were involved in a different set of patients with unique PMI's and stage of pathology.
2. No details on PMI or RIN values for each case unless buried deep in sup data. This would be good to have as a co variant.

3. It would be useful to compare male and female cases separately given different disease progression between sexes. The FTD had one more male than female
4. In many cases the results are over interpreted without further studies. For example the results “suggests cell-autonomous initiation in ALS (including C9-ALS) specifically in glutamatergic neurons” and “a convergence of molecular pathology in C9-ALS excitatory neurons on protein misfolding, which triggers mitochondrial dysfunction and impairment of translation as common endpoints associated with neurodegeneration.”
5. Seems like they have enough cells in Figure 1B to further breakdown the deep layer and intermediate layer cell types to analyze higher resolution, rather than just three major neuronal types
6. The fact that they could not see Betz cells – the primary cell type leading to paralysis in ALS was rather worrying but of course this is difficult given how rare these cells are in the motor cortex. This should be discussed more.
7. More confirmation of key findings with follow up analysis using western or immune would significantly strengthen their discoveries and add to the paper.

Response to reviewers' comments

Reviewer #1 (Remarks to the Author):

Li et al provide a single cell omic analysis of C9orf72 ALS/FTD and healthy controls, including transcriptomics, ATACseq and H3K27Ac CHIP. snRNAseq from ALS is by far the most robust dataset, because the C9-FTD samples appear to have insufficient quality for a full analysis (neuron data is missing). Some hits are validated by immunoblotting. Comparing frontal vs motor cortex makes a lot of sense, although the results are surprisingly similar. The epigenetic methods are ambitious but provide limited additional information, because they seem statistically underpowered. Nevertheless, the manuscripts will be a valuable resource for the community and should be published after revision.

Major points:

-From the abstract it is unclear to what the C9orf72-ALS and FTD cases are compared. To each other, to non-C9-ALS/FTD or to healthy controls? This is answered early in the paper, but the title and abstract insinuate C9-specific effects are analyzed, which is not the case, because no ALS/FTD cases without the mutation were analyzed. The nature of the controls becomes clear early in the text but the limitations are not mentioned anyway, e.g. effects on translation are directly attributed to DPRs, although the same changes may occur in non-C9-ALS/FTD. Probably the majority of the identified changes are also present in sporadic ALS/FTD, which has very similar clinical presentation. See competing preprint (<https://www.biorxiv.org/content/10.1101/2021.07.07.451374v2.full>). While it makes sense to focus on C9orf72 because they are potentially more homogenous than sporadic ALS and FTD, but without comparison to sporadic ALS/FTD the interpretation will also be limited. Without such new data the title should be revised.

We fully agree with these comments. In the Abstract, we have now clarified that in addition to C9-ALS and C9-FTD cases, we also analyzed a control group of donors. We revised the title of the paper to avoid the ambiguity pointed out by the Reviewer. The new title, "*Divergent impacts of ALS and FTD on neurons and glia in patients with C9orf72 mutation*", describes our experiments and findings more accurately.

-There is significant overlap with a preprint that actually compared C9orf72 vs sporadic ALS/FTD in a larger cohort (<https://www.biorxiv.org/content/10.1101/2021.07.07.451374v2.full>), which should be discussed and analyzed based on the data available on biorxiv. The present manuscript uses far less sample but also looks at epigenetics.

We are aware of the preprint by Pineda *et al.*, and we have carefully studied the findings reported there. We find some consistencies but also differences between our dataset and the findings reported by Pineda *et al.*. However, given that the preprint has not been peer-reviewed or published, and that some of the underlying data is not publicly available, we are not able to thoroughly compare our findings and interpret the potential similarities and differences. We look

forward to the opportunity to compare our findings in detail when the Pineda *et al.* paper is published.

-Fig 2G: Since the control and C9-ALS samples differ by genotype and disease status it is impossible to tell whether the C9orf72 repeat expansion triggers a cell type specific reduction in C9orf72 expression or whether a general suppression is overlaid by ALS-specific upregulation, e.g. in microglia.

We agree with the Reviewer that our data cannot distinguish a direct effect of C9orf72 mutation vs other effects related to the diseases. We have added a clarifying sentence to the Results section, "Disruption of gene expression in C9-ALS astrocytes and excitatory neurons,":

"...However, because C9-ALS and control samples differ by both genotype and disease status, our study cannot determine whether downregulation of *C9orf72* expression in specific cell types is directly caused by the repeat expansion mutation, or is also influenced by ALS disease processes."

-Validate GOLGB1 and especially MAOB by western blot, because large part of the discussion is based on it.

-Validate mitochondrial hits by western blot given conflicting results showing reduction of mitochondria in ALS patients using EM.

To address the Reviewer's comments, we used Automated Western in bulk tissue from motor cortex to analyze MAOB and GOLGB1 proteins, which are encoded by genes that were DE in astrocytes, as well as several mitochondrial and ribosomal proteins, which are encoded by genes that were DE in excitatory neurons. In total, we tested 10 additional proteins using 21 different antibodies. Antibodies against 4 of these 10 proteins produced a protein band of the expected size in control samples (*MAOB*, *ATP5A*, *CYCS*, *COX5A*). However, these experiments did not confirm the predicted changes at the protein level for these DE four genes.

Specifically, we tested two different antibodies for GOLGB1 (Giantin), as well as several different antibodies for ribosomal proteins RPL28 and RPS29. None of these antibodies produced a protein band of the expected size. We obtained a band of the expected size for MAOB using 2 different antibodies but did not detect a difference in MAOB protein levels between C9-ALS and controls (Extended Data Fig. 6B-C; n=9 cases and 5 controls; two repeated experiments; p=0.549 and 0.297, two-sided Welch's t-test).

B**C**
Extended Data Fig.6. (B) MAOB protein levels from nine C9-ALS and five control donors. Each sample was assessed in two repeated experiments. (C) Quantifications of protein levels in (B). Circles represent the signal for each donor. For each repeat experiment, a two-sided Welch's t-test was used to compare the immunoreactive signals between C9-ALS and control. No significant changes of MAOB protein levels were detected ($p=0.549$ and 0.297).

We tested multiple antibodies against the products of 5 differentially expressed mitochondrial genes that were upregulated in C9-ALS excitatory neurons (*ATP5A*, *CYCS*, *UQCRH*, *NDUFA4*, *COX5A*). We obtained a protein band of the expected size for 3 of them (*ATP5A*, *CYCS*, *COX5A*). The protein levels of *ATP5A*, *CYCS*, and *COX5A* did not differ between C9-ALS and controls ($n=10$ cases and 6 controls for *CYCS*; $n=9$ cases and 5 controls for *ATP5A* and

COX5A; two repeated experiments; $p=0.206-0.757$, two-sided Welch's t-test; see Extended Data Fig.7C-D).

Extended Data Fig.7. (C) ATP5A, CYCS and COX5A protein levels ($n=10$ C9-ALS and 6 control donors for CYCS; $n=9$ C9-ALS and 5 control donors for ATP5A and COX5A). Each sample was assessed in two repeated experiments. (D) Quantifications of protein levels in (C). Circles represent the signal for each donor. For each repeat experiment, a two-sided Welch's t-test was used to compare the immunoreactive signals between C9-ALS and control. No significant changes of ATP5A, CYCS or COX5A protein levels were detected ($p=0.206-0.757$).

We note that, in addition to astrocytes, MAOB is expressed in neurons and several glial cell types, and mitochondrial genes are expressed in all cell types. Our protein assays were performed in bulk brain tissue, precluding the detection of the astrocyte- or neuron-specific signals. Although it is theoretically possible to obtain quantitative cell-type-specific signals in human postmortem brain using IHC or IF, these experiments require multiplex staining with at least 3 different antibodies and DAPI. High-quality tissues are needed for this work, and we did not have adequate specimens to perform these experiments. Therefore, we could not reliably evaluate the relationship between transcriptional and protein alterations for these targets in C9-ALS astrocytes or neurons.

To avoid overstatements, we have modified the Results and Discussion section of the revised manuscript, deleting or modifying the statements regarding GOLGB1, MAOB, and mitochondrial genes. We have also removed GOLGB1 from Fig. 3A-B.

-Fig 6B: Is neuron loss really that much more severe in FTD than in ALS? Quantify neuron loss in samples by other means, e.g. IHC or flow cytometry.

We evaluated the proportion of nuclei from neurons in each sample by multiple means. Using the FANS sorting data (Extended Data Fig.5A), we found that the median proportion of NeuN+SOX10- nuclei (putative neurons) was consistent (~25-37%) across each of the donor groups (Extended Data Fig.5D and updated Table S1; no significant differences in pairwise comparisons of C9-ALS vs. Control or C9-FTD vs. Control, p values=0.41-0.97 by t-test). We also analyzed the single nuclei RNA-seq data using relaxed QC criteria (without thresholds on the number of detected genes, the percentage of mitochondrial reads, and doublet scores). This analysis included all non-empty droplets that were identified by CellBender. We found that ~36-50% (median per group) of the droplets contained putative neurons as determined by clustering and expression of neuronal marker genes (e.g. *SNAP25*, *MEF2C*), and that there were no significant differences between control and C9-ALS or C9-FTD groups (Extended Data Fig.12A; t-test p-values=0.20-0.76). We, however, detected that a larger proportion of neurons were filtered out by our QC criteria in C9-FTD vs. C9-ALS or control samples, especially in frontal cortex (Fig. 6B). The filtered cells had a lower number of detected genes and a higher proportion of mitochondrial RNA reads (Extended data Fig.1A). Taken together, these complementary data indicate that, although neurons may be equally prevalent in C9-FTD compared with control or C9-ALS brain samples, they contained a lower amount of detectable nuclear RNA.

An important caveat of our data is that we cannot distinguish between lower tissue quality due to technical factors (e.g. tissue handling) vs. disease effects. However, it is striking that the lower proportion of high-quality nuclei that we observed is limited to a specific cell type (neurons), donor population (C9-FTD), and brain region (frontal cortex), which is consistent with the expected pattern of tissue-level pathology in C9-FTD. If these effects were caused by sample quality, we would expect a more general pattern affecting all cell types and brain regions.

To clarify these findings, we have included the cell proportion from FANS in Table S1, modified the main text, added new panels in Extended Data Fig.5D and Extended Data Fig.12A-B, and rearranged panels accordingly.

The section, “A distinct signature of glial dysregulation in C9-FTD,” was modified as follows:

“...Notably, the proportion of high-quality excitatory and inhibitory neurons was 2.1-fold lower in frontal cortex in C9-FTD compared with control (**Fig. 6A-B**). No such reduction in the proportion of neurons was observed in C9-ALS. The reduction in high-quality neurons in C9-FTD samples was mainly due to a high proportion of low-quality nuclei with a smaller number of detected genes and a high proportion of mitochondrial RNA (**Extended Data Fig. 1A-C; Extended Data Fig. 12A**). Notably, flow cytometry showed no difference in the overall proportion of neurons (NeuN+ SOX10- cells) in any of the donor groups (**Extended Data Fig. 5D**), suggesting that the depletion of the frontal

cortex C9-FTD neurons in snRNA-seq was due to the lower quality of neuronal nuclear RNA. Neurons were also depleted among high-quality snRNA-seq nuclei in many of the motor cortical samples, but the proportion was more variable and not significantly different from controls on average (**Fig. 6B**). Among neurons, the ratio of excitatory to inhibitory neurons was not significantly different (**Fig. 6C; Extended Data Fig. 12B**). Although our study cannot distinguish between lower tissue quality due to technical factors (e.g., tissue handling) vs. disease effects, the lower proportion of high-quality nuclei that we observed was limited to a specific cell type (neurons), donor population (C9-FTD), and brain region (frontal cortex), which is consistent with the expected pattern of tissue-level pathology in C9-FTD.”

Extended Data Fig.12. (A-B) Related to Fig. 6B-C. Relative abundance of neurons (A), and percent of excitatory neurons among all neurons (B) using relaxed QC criteria (without thresholds on the number of detected genes, the percentage of mitochondrial reads, and doublet scores). This analysis included all 195,144 non-empty droplets determined by CellBender(Fleming et al. 2019). No significant difference in the percentage of neurons was detected between diagnostic groups in (A) (t-test, $p=0.20-0.76$). Circles represent individual donors. *, t-test $p < 0.05$.

Extended Data Fig.5D. Percentage of putative neurons (NeuN+ SOX10- nuclei) from FANS. Circles represent individual donors. No significant differences were found in any pairwise comparison between diagnostic groups (t-test, $p=0.41-0.97$).

We have also changed the title of Fig. 6:

Fig. 6: Lower proportion of high-quality neurons and alteration of non-neuronal transcription in C9-FTD.

-Elaborate on differences ALS and FTD and discuss selected genes

Thank you for this suggestion. In response to this comment, we added Fig. 6I (see below) and provided examples of genes that show different patterns of expression changes in C9-ALS and C9-FTD.

Fig.6 (I) Comparison of the disease fold-changes in C9-ALS and C9-FTD for astrocytes (Astro), oligodendrocytes (Oligo), and OPC. To avoid double-dipping, the control donors were split into two groups and used to compute the fold-changes for C9-ALS and C9-FTD respectively (see Methods). Only genes that were significantly DE in both comparisons with these split controls are shown in the scatter plot. Highlighted are examples of genes that are also significantly DE in our full model reported in Fig.2A and Fig.6D.

We added the following to the last paragraph of the section, “A distinct signature of glial dysregulation in C9-FTD”:

“...To identify the most pronounced differences, we looked at genes that were DE in both C9-ALS and C9-FTD, and highlighted several genes that were altered in the same or in opposite directions (**Fig. 6I**). We found that *GFAP* was strongly upregulated in motor cortex in both diseases, consistent with activation of astrocytes. On the other hand, the

voltage-gated potassium channel regulator protein, *LG1*, was upregulated in C9-FTD but downregulated in C9-ALS in astrocytes from motor cortex. Notably, clearance of extracellular potassium by astrocytes is essential for controlling neuronal excitability. *NAV2* that encodes Neuron Navigator 2, showed an opposite pattern of expression in oligodendrocytes from frontal cortex. Neuron Navigator 2 was suggested to be involved in CNS development, and the *NAV2* gene has been implicated in Alzheimer's Disease.”

Minor points:

-What do the box plot next to the GO dotplots mean? e.g. Fig. S8A

We described the methods that we used for this analysis in the second paragraph of the “Gene Ontology (GO) enrichment test” section of the Material and Methods, as well as in the figure legends for Fig.4D and Extended Data Fig.8A. In short, these boxplots show the difference in the effect of the disease (log fold change in C9-ALS vs. Control) between motor and frontal cortices ($\Delta \log_2FC = \log_2FC$ in motor cortex - \log_2FC in frontal cortex). The boxes denote the distribution of these differences ($\Delta \log_2FC$) for up-regulated DE genes (in Fig.4D) or down-regulated DE genes (in Extend Data Fig.8A) for each GO category. As background comparisons, we also show the distributions of these differences for all expressed genes (labeled as “all genes”) and for all ALS vs. Control DE genes (labeled as “DE gene”) on top.

We now clarify the figure legends as follows:

Fig.4 (D) Differences in the C9-ALS vs. Control fold-changes between motor and frontal cortex ($\Delta \log_2FC = \log_2FC$ in motor cortex - \log_2FC in frontal cortex). The boxes denote the distribution of these differences ($\Delta \log_2FC$) for C9-ALS vs. Control up-regulated DE genes for each GO category. As background comparisons, the distributions of these differences for all expressed genes (labeled as “all genes”) and for all ALS vs. Control DE genes (labeled as “DE gene”) are shown on the top. T-test was used to test whether the $\Delta \log_2FC$ in each group of genes were significantly different from the $\Delta \log_2FC$ of the “all genes” control set, and asterisks (*) mark the significant differences (FDR<0.001).

Extended Data Fig.8 (A) *Left panel*: Top gene ontology (GO) terms enriched for genes downregulated in upper and/or deep layer excitatory neurons. Enrichments of the same terms for downregulated DE genes in other neuronal cell types are shown for comparison. Enriched GO categories (FDR<0.01) were selected by affinity propagation. *Right panel*: Differences in the C9-ALS vs. Control fold-changes between motor and frontal cortex ($\Delta \log_2FC = \log_2FC$ in motor cortex - \log_2FC in frontal cortex). The boxes denote the distribution of these differences ($\Delta \log_2FC$) for C9-ALS vs. Control down-regulated DE genes for each GO category. As background comparisons, the distributions of these differences for all expressed genes (labeled as “all genes”) and for all ALS vs. Control DE genes (labeled as “DE gene”) are shown on the top. T-test was used to test whether the $\Delta \log_2FC$ in each group of genes were significantly different from the $\Delta \log_2FC$ of the “all genes” control set. No significant differences were found.

-Even though the clustering seems thorough and valid, it is unclear how the authors determined the clustering resolution. Usage of clustree is highly recommended.

Thank you for this suggestion. We have applied clustree analysis to visualize the stability of the clusters as a function of the Leiden resolution parameter (see below). This analysis shows that the cluster resolutions we have used before annotations (circled in red) correspond to relatively fine-grained, somewhat over-clustered cell populations. We subsequently merged these clusters based on the expression of marker genes reported by the Allen Institute (Bakken *et al.*, as described in Methods and Extended Data Fig.3-4), which increases the robustness of the clusters. Our goal was to analyze the major cell types corresponding to previously reported and well-characterized human brain cell populations, not to push the cluster resolution to its limit.

We have now included this new analysis in Extended Data Fig. 2F and updated the main text and the methods as follows:

In the “Single nucleus transcriptomes in C9-ALS and C9-FTD specimens identify fine-grained cortical cell types” section of the Results:

“...The clustering was not biased by brain region, donor sex, sequencing batch, individual samples, or diagnosis (Fig. 1C, Extended Data Fig. 2A-E), and the resulting clusters were stable with respect to the resolution parameter for community detection (Extended Data Fig. 2F).”

In the “snRNA-seq data preprocessing, clustering and cell type annotation” section of the Materials and Methods:

“...This iterative clustering process reselects the highly variable genes in the context of each cell class and hence may give better resolution in finding distinct subclusters. We used SC3 stability index and Clustree to evaluate and visualize the stability of the clusters across a range of resolutions for the Leiden community detection analysis (Extended Data Fig.2F). With the Leiden resolution set as 1.5, 0.5 and 1, we identified 24, 23, and 23 subclusters for excitatory neurons, inhibitory neurons, and non-neurons, respectively.”

Extended Data Fig. 2. (F) Clustree analysis to evaluate cluster stability with increasing Leiden resolutions from 0 to 3 in our full dataset clustering and the sub-clusterings for the three major cell classes. Clusters found at each resolution value are represented by nodes in the corresponding column. The nodes are colored by the SC3 stability index and sized proportional to the number of nuclei in the cluster. The transparency of the edges is adjusted according to the in-proportion, a metric defined as the ratio between the number of nuclei on the edge and the number of nuclei in the cluster it goes toward. Low in-proportion edges tend to arise as clusters become unstable. The selected resolutions we used before cell type annotation are highlighted in red circles.

-Please provide specific version of Seurat that were used for this analysis. More information is promised in supplementary methods, but I could not find it.

Seurat (v4.0.4) was used. This is stated in the "snRNA-seq data preprocessing, clustering and cell type annotation" part of the Material and Methods section.

Reviewer #2 (Remarks to the Author):

The authors describe an extensive single-nucleus-based transcriptomic and epigenomic characterization of post-mortem motor and frontal cortices in C9-ALS and C9-FTD cases. The data are of high quality and the analyses are competent. There are some intriguing findings that suggest the different types of neural cells may be differentially affected in different regions of the brain, and that this may depend on the clinical and pathological presentation of the disease. The authors go on to discuss the implications of these findings. Overall, this would be a valuable resource to the community, although the studies could be strengthened further.

Thank you for recognizing our study's value and the quality of our data and analyses.

Key concerns include the following:

1. In the absence of comparisons with other neurodegenerative diseases, it is difficult to see how many of these findings are specific to ALS-FTD and how many are general responses to a degenerative process (e.g., changes in astrocytes – are the changes described specific to this disease?).

We thank the reviewer for this suggestion. We compared our DE genes (C9-ALS vs. Control) with the reported DE genes from a recent multiome single nucleus study of Alzheimer's disease (AD) in prefrontal cortex (Morabito *et al. Nat Genet.* 2021). Overall, we found a significant but low overlap between our C9-ALS DE genes and the AD DE genes (the Jaccard index of overlap < 0.14 for all comparisons; see below and the new Fig.2H). The overlap was highest for glial cell types (astrocytes and oligodendrocytes), suggesting a shared component of transcriptional disruption in glia across these two neurodegenerative diseases, whereas the impact of the diseases on neurons was more distinct. For genes that were significantly DE in both studies, the fold-changes were in general correlated between the two diseases ($r > 0.58$, $p < 10E-15$ for astrocytes and oligodendrocytes, and $r > 0.23$, $p < 0.006$ for neuronal subtypes in prefrontal cortex; see below and the new Extended Data Fig. 13A). We also examined the significant DE genes that had concordant or discordant effects in ALS (our study) and AD (Morabito *et al.*). For this comparison, we focused on cell types that were most affected in C9-ALS, i.e. astrocytes, Exc upper/deep neurons, as well as oligodendrocytes that had the highest Jaccard index (see below and the new Extended Data Fig.13B-E). We found that in both diseases, *CD44* and *MAOB* were upregulated in astrocytes, whereas *DNMT3A* was downregulated and *HSPH1* was upregulated in both neuronal subtypes. In contrast, *KNDC1* was upregulated in C9-ALS but downregulated in AD in the prefrontal cortex astrocytes, whereas *HTR1E* was downregulated in C9-ALS but upregulated in AD in the prefrontal cortex deep excitatory neurons.

We have added these findings to Fig.2H and Extend Data Fig.13, as well as the following text to the "Disruption of gene expression in C9-ALS astrocytes and excitatory neurons" section of the Results:

"To examine whether our findings were specific to C9-ALS or were general responses to a degenerative process, we compared the DE genes in our C9-ALS vs. Control analysis with the reported DE genes in Alzheimer's disease (AD) in prefrontal cortex. We found little overlap between the C9-ALS DE genes and the AD DE genes (the Jaccard index < 0.14 for all comparisons) (**Fig. 2H**). The overlap was highest for glial cell types (astrocytes and oligodendrocytes), suggesting a shared component of transcriptional disruption in glia across these two neurodegenerative diseases. The impact of C9-ALS and AD on neurons was more distinct. For genes that were significantly DE in both diseases, the fold-changes were in general correlated between the two diseases ($r > 0.58$, $p < 1E-14$ for astrocytes and oligodendrocytes, and $r > 0.23$, $p < 0.006$ for neuronal subtypes in prefrontal cortex) (**Extended Data Fig. 13A**). We also examined the significant DE genes that had concordant or discordant effects in ALS (our study) vs.

AD³⁰. For this comparison, we focused on the cell types that were most affected in C9-ALS, such as astrocytes and excitatory upper/deep neurons, as well as oligodendrocytes that had the highest Jaccard index (**Extended Data Fig. 13B-E**). We found that *CD44* and *MAOB* were upregulated in astrocytes in both C9-ALS and AD, whereas *DNMT3A* was downregulated and *HSPH1* was upregulated in both neuronal subtypes. In contrast, *KNDC1* was upregulated in C9-ALS but downregulated in AD in prefrontal cortex astrocytes, whereas *HTR1E* was downregulated in C9-ALS but upregulated in AD in the prefrontal cortex deep excitatory neurons.”

Fig.2 (H) Overlaps of significant C9-ALS vs. Control DE genes (this study) and AD vs. Control DE genes (Morabito *et al.*) for each cell type, as measured in Jaccard index.

Extended Data Fig. 13. **Comparison of C9-ALS vs. Control gene expression fold-changes with AD vs. Control gene expression fold-changes.** (A) Spearman correlations of disease fold-changes between C9-ALS (this study) and AD (Morabito *et al.* 2021) for matching cell types. Only genes that were significantly DE in both diseases were included in this analysis. Asterisk denotes significant correlations

(FDR < 0.05). (B-E) Scatter plots showing the correlation of the disease fold-changes between the two diseases for astrocytes, oligodendrocytes, and excitatory neurons in the two studies. Examples of genes with concordant or discordant changes in the two diseases are highlighted. n, number of significant DE genes in both studies in the corresponding cell types; rho and p, Spearman's correlation coefficient and p-value.

2. Were the C9-ALS cases specifically assessed for cognitive dysfunction and was it confirmed to be absent? Were these cases also assessed post-mortem to confirm absence of pathology in non-motor regions? It would be helpful to clarify this in the manuscript (apologies if the authors have done so and I missed it).

In response to this comment, we added information to Table S1, as well as clarified the cognitive status and neuropathological assessment of our cases in the Human Samples section of the revised manuscript.

In short, approximately 50% of all ALS patients exhibit cognitive or behavioral symptoms of FTD, and 15% to 41% of ALS patients present cognitive impairment (Lomen-Hoerth, 2011; Abramzon et al. 2020). Notably, C9-ALS carriers have distinct characteristics, including a younger age at onset, a higher proportion of concomitant FTD, a higher prevalence of psychotic symptoms and cognitive impairment, and a worse prognosis (Abramzon et al., 2020; Glasmacher et al., 2020). Characteristically, motor and cognitive phenotypes range from pure ALS to pure FTD with a large spectrum of intermediate phenotypes (Zampatti et al., 2022). Typically, ALS clinical features involve motor impairment with some symptoms of cognitive decline and/or behavioral changes (Zampatti et al., 2022).

The case selection for our study was performed by the expert neuropathologist (Dr. Dennis W Dickson). ALS/FTD neuropathologic changes were assessed in both BA4 (primary motor cortex) and BA9 (DLPFC, hereafter PFC) using immunohistochemistry for TAR DNA binding protein 43 (TDP-43). C9-FTD cases presented TDP pathology in PFC with almost no inclusions in motor cortex, along with no Betz cell depletion. In C9-ALS cases, TDP pathology was concentrated in motor cortex with a severe depletion of Betz cells. Other brain regions, such as hippocampus, basal forebrain, basal ganglia, thalamus, midbrain, pons, medulla, cerebellum were also evaluated using multiple markers to exclude other neurodegenerative pathologies.

The selected C9-ALS cases had clinical features of muscle weakness and atrophy with fasciculations, with at most mild cognitive and memory impairment (3 of 10 cases) or mild dementia (2 of 10). None of the selected C9-ALS cases had parkinsonism or significant apathy. Only 1 case had mild depression and 2 cases had some personality changes. In contrast, all the 7 FTD cases had significant dementia with behavioral abnormalities (most often frontal type), but only 2 had mild muscle weakness.

The following was added to the "Human Samples" section in Materials and Methods:

“...ALS/FTD neuropathologic changes were assessed in both BA4 and BA9 using immunohistochemistry for TAR DNA binding protein 43 (TDP-43). All C9-ALS cases had clinical features of muscle weakness and atrophy with fasciculations, with at most mild cognitive and memory impairment (3 of 10 cases) or mild dementia (2 of 10 cases) (**Table S1**). None of the C9-ALS cases had parkinsonism or significant apathy. Only 1 case had mild depression and 2 cases had some personality changes. In contrast, all 7 FTD cases had dementia with behavioral abnormalities (most often frontal type), but only 2 had mild muscle weakness.”

3. If possible, it would be helpful to test some of the speculations in the discussion on post-mortem tissue – e.g., would you expect to see an upregulation of translation system components close to where there is evidence of DPRs, or an upregulation of ROS scavengers in neurons with signs of DNA damage? Could this be tested by IF experiments? If yes, is it generalizable or specific to ALS-FTD?

IHC or IF experiments in human postmortem tissue are not amenable for testing spatial relationships between different pathological pathways in a quantitative and cell-type-specific manner. Our validation of snRNA-seq findings was therefore mostly restricted to quantitative analysis using the automated Western blotting in bulk brain tissue (see also responses to Reviewer #1). We tested commercially available antibodies against DRPs [Anti-C9ORF72/C9RANT (Poly-PR) Cat ABN1354 and Anti-C9ORF72/C9RANT (Poly-GR) Cat ABN1361; Sigma]. These antibodies were previously validated in cell culture or iPS-derived neurons but had not been tested in human postmortem brain. The antibodies did not yield bands of the expected sizes in our experiments.

We also tested antibodies for SOD1 and found no changes in the expression of the SOD1 protein between C9-ALS and controls in bulk cortical tissue (Extended Data Fig. 7). There could be multiple explanations for this discrepancy. Transcriptional alterations that we detected might not result in changes in the abundance of the SOD1 protein. Alternatively, SOD1 is highly expressed in multiple brain cell types, precluding the detection of the signal that is specific for neurons in bulk tissue. To avoid speculations, we removed the statement about the effect of ROS from the Discussion section of the paper.

Minor points:

1. The transcriptomic and epigenomic data are analyzed separately – would it be possible to assess them together, to potentially highlight transcriptional regulatory circuits?

Thank you for this suggestion. Indeed, we tried to jointly analyze the data modalities by multimodal integration. However, because of the sparsity of the snATAC-seq data (i.e. relatively few reads sequenced per cell), it was not possible to cleanly integrate both datasets with appropriate cell type resolution. We believe this is a major challenge for single-cell epigenomic assays (see e.g. Lee et al. Biorxiv 2023 <https://www.biorxiv.org/content/10.1101/2023.02.01.526609v1.full>), which could be addressed in

the future by multi-modal techniques such as 10X multiome which measure both transcriptome and chromatin accessibility in the same cell. Given the current dataset, our strategy was first to cluster the snATAC nuclei only using the open chromatin information and later annotate them based on CCA integration with our snRNA dataset (see Methods), and then focused on the disease effect (fold-changes) in the two modalities. In Fig. 5 we show that there is a significant correlation between the effect of ALS on the epigenome and transcriptome.

2. There is a lot of descriptive data here, which I found difficult to keep track of – it would be helpful if there were a schematic/graphical/cartoon representation of the various cell- and layer-specific changes in the different cortical regions, and their relationship to the clinical and pathological presentation of the disease. It would help the readers to better absorb and appreciate the data.

We thank the reviewer for the suggestion. We now included the following schematic in Fig.7:

Fig 7. Summary of the major results of our snRNA-seq analysis of postmortem brain samples from C9-ALS donors. We detected that the most pronounced transcriptional disruption in C9-ALS concentrated in excitatory neurons and astrocytes. These changes were highly consistent in motor and frontal cortices (not shown). C9-ALS astrocytes showed increased expression of genes associated with activation and structural remodeling. C9-ALS upper-layer (L2/3) and deep-layer (L5/6) excitatory neurons had increased

expression of genes related to proteostasis, metabolism, whereas genes related to neuronal function were down-regulated. In both cortical regions, there were more extensive gene dysregulation in upper layer vs. deep-layer excitatory neurons (not shown).

Reviewer #3 (Remarks to the Author):

This is a thorough analysis of the frontal and motor cortex in patients with C9ORF72 mutations displaying either FTD (n=5) or ALS (n=6) compared to control cases (n=6). The patient groups were generally well matched for age, sex and race. This is certainly a fantastic resource paper but not a primary research paper as there is no validation of any of their results or mechanistic insight into the snSEQ analysis. This also leads to another problem which is that many of the results do not support their conclusions without lack and gain of function experiments. In fact the authors say in the caveats section “we cannot distinguish cell-type-specific alterations that represent direct effects of C9 mutation from end-of-life consequences of neurodegeneration”. And yet in the paper they do make assumptions from this data set. This was the major weakness of the paper. But the data have been generated in a very reliable way and would be of great value to the community.

Thank you for the kind appreciation of the value of our study, and for your helpful critiques which we address below.

1. The major weakness of this paper is that the authors move directly to supervised statistical methods to look for DEGS between C9 and control in specific genes/cells rather than a pseudobulk PCA analysis to see if ALS and control reliably separate without prior labeling. The history of this type of analysis is that there will always be DEGS in a single data set that are significant when looking at this many groups of cells and numbers of genes. While FDR was used this is prone to issues across this many cell types and genes. The gold standard would be to run a validation set of samples to see if the same gene sets were involved in a different set of patients with unique PMI's and stage of pathology.

We understand the reviewer's concern. We agree that a gold standard validation experiment on a new set of samples would be the most rigorous validation, but it is unfortunately beyond the scope of our study. Instead, we have taken multiple approaches to address this issue through detailed computational analyses.

First, as recommended by the reviewer we performed unsupervised analysis to characterize the similarity of samples from each patient in an unbiased manner. We created pseudobulk counts with nuclei originating from each donor for each cell type, computed normalized counts with DESeq2's median of ratios method, and performed PCA separately for each cell type and each brain region using the log normalized counts (see Fig. R1 below). Although the number of subjects is limited, we do observe a separation of the disease and control samples, especially in

astrocytes and excitatory neurons. The inter-individual variability of gene expression was greater in some cell types than others.

A**B**

Fig.R1. Principal component analysis of the pseudobulk counts from each donor in each cell type. (A-B) Separation of the pseudobulk gene expression profiles in each major cell type for ALS and Control donors in motor cortex (A) and frontal cortex (B). (C-D) Separation of the pseudobulk gene expression profiles in each non-neuronal nuclei for FTD and Control donors in motor cortex (C) and frontal cortex (D). Counts were normalized with DESeq2's median of ratios method, and the natural logarithm of the normalized count plus one was used in the PCA analysis.

Second, we used Augur (Skinnider *et al.*, *Nat. Biotechnol* 2021) to quantify the separability of nuclei from different diagnoses. Augur employs a machine-learning framework to quantify the separability within a high-dimensional space. For each cell type, Augur withholds a proportion of sample labels and trains a random forest classifier on the labeled subset. The classifier predictions are compared with the experimental labels, and cell types are prioritized on the basis of the area under the receiver operating characteristic (AUC) of these predictions in cross-validation. This analysis showed that we can distinguish nuclei from disease or control samples using the gene expression profile, with the highest AUC of ~0.78 achieved in astrocytes for ALS vs. Control (see Fig.R2A below). Furthermore, for the ALS vs. Control analysis, we generated random shuffles to check how much better these observed AUCs would perform than the shuffles. We permuted the diagnosis labels of the individual donors (so that all nuclei from each donor will still have the same diagnosis labels), and required that the two shuffled groups must each contain three C9-ALS donors and three Control donors, of which three are male and three are female. This resulted in 82 possible permutations, and we ran Augur to get the AUCs for distinguishing the two groups. We found that in motor cortex the AUCs from the real diagnosis labels are higher than the 95% upper confidence intervals in astrocytes, excitatory neurons and VIP inhibitory neurons. Astrocytes and deep-layer excitatory neurons also have higher AUCs in the observed than those in the shuffles in the frontal cortex samples (see Fig.R2B below).

Fig.R2. Cell type prioritization using Augur. (A) The area under the receiver operating characteristic curve (AUC) was reported by Augur as a metric of the separability between the disease and control samples. (B) For ALS vs. Control, 82 random shuffles with balanced diagnosis and sex were generated by permutation of the diagnosis labels of the donors. Augur was run on these shuffles and the mean AUCs are shown in grey dots. Error bars represent 95% confidence intervals of the mean. Observed AUCs with real diagnosis labels are shown in red dots. Cell types that didn't show up in all 12 donors were omitted in this analysis.

Last, we applied linear discriminant analysis (see Fig.R3 below) to quantify the discriminability of C9-ALS cells vs. Control cells. We separated the data into training and test sets based on the donor, using 10 donors for training and 2 for testing (one each from C9-ALS and Control). We trained a LDA classifier using the top 1-10 principal components for each cell type and brain region. The results showed that C9-ALS cells could be meaningfully distinguished from Control using the top few principal components in deep and superficial Exc neurons, and in astrocytes.

Fig.R3. Discriminability of C9-ALS vs. Control cells is shown as the area under the receiver operating characteristic for four sets of cells. Donors were split using k-fold cross-validation, with 10 subjects used for training a linear discriminant classifier and 2 subjects used for testing (one control and one C9-ALS subject). The AUC for training subjects (orange) increases with the number of PCs used, whereas the test-set AUC (blue) shows that the C9-ALS cells can be significantly discriminated from Control using the top few PCs.

Altogether, we hope the reviewer will appreciate that despite the presence of inter-individual variability, our data demonstrate a significant ability to distinguish between disease and control samples. This distinction is particularly pronounced in the case of astrocytes and excitatory neurons, which have been the primary focus of our research.

2. No details on PMI or RIN values for each case unless buried deep in sup data. This would be good to have as a covariant.

The Mayo Clinic team provided us with all available information for the donors used in the study, and we added this information to the revised supplementary Table S1. Unfortunately, PMI was available for only ~50% of donors, and RINs were not assessed in the bulk brain tissues of any donors.

3. It would be useful to compare male and female cases separately given different disease progression between sexes. The FTD had one more male than female.

Thank you for the suggestion. We tried analyzing the male and female subjects separately, and found there was a very significant overlap in the group of DE genes called in the two sexes. However, because we have too few subjects of each sex to adequately power this analysis we have not further investigated sex differences in this study.

4. In many cases the results are over interpreted without further studies. For example the results “suggests cell-autonomous initiation in ALS (including C9-ALS) specifically in glutamatergic neurons” and “a convergence of molecular pathology in C9-ALS excitatory neurons on protein misfolding, which triggers mitochondrial dysfunction and impairment of translation as common endpoints associated with neurodegeneration.”

We agree with this comment, and we removed this and several similar statements from the revised manuscript.

5. Seems like they have enough cells in Figure 1B to further breakdown the deep layer and intermediate layer cell types to analyze higher resolution, rather than just three major neuronal types

We agree that our data could be analyzed at a finer resolution (see the response to Reviewer 2, clustree analysis; also see our rationale in the second paragraph of the “Single nucleus transcriptomes in C9-ALS and C9-FTD specimens identify fine-grained cortical cell types” in the Results section). However, our goal was not to achieve the maximum possible fine-grained cell-type resolution. Instead, we analyzed the data at the level of major cell populations to ensure that the details of clustering would not have a major impact on the analysis of subtle disease-related changes in gene expression. The level of cluster resolution we used is similar to those used in other single-cell studies of neurodegenerative disease. Given the limited number of donors in our study, even though we do have many cells sampled from each donor, we felt it would be imprudent to analyze disease effects in fine-grained excitatory neuron subtypes.

6. The fact that they could not see Betz cells – the primary cell type leading to paralysis in ALS was rather worrying but of course this is difficult given how rare these cells are in the motor cortex. This should be discussed more.

We elaborated on this issue in the revised paper, providing more information about the difficulties in reliably identifying Betz cells in single-nucleus RNA-seq studies. We added the following in the Discussion section:

“Anatomical studies showed that ALS causes degeneration of the giant upper motor neurons (Betz cells) which are located in layer 5b of the primary motor cortex and are relatively rare cells that account for ~10% of the total number of pyramidal cells in layer 5b (Rivara et al. 2003). A recent large-scale study of the human motor cortex identified

cells consistent with the size and shape of Betz cells in two Layer 5 extratelencephalic clusters (Exc L3-5 *FEZF2 ASGR2* and Exc L5 *FEZF2 CSN1S1*), but these clusters also included other neurons with pyramidal morphologies (Bakken et al. 2022). In the absence of specific markers, we were not able to reliably separate the cluster that corresponds to Betz cells in our snRNA-seq data. However, all deep layer (L5/6) excitatory neurons are generally considered vulnerable in ALS (Maekawa et al. 2004)(Nolan et al. 2020), and we found transcriptional dysregulation of L5/6 excitatory neurons which was more extensive in motor than frontal cortex.”

7. More confirmation of key findings with follow up analysis using western or immune would significantly strengthen their discoveries and add to the paper.

Please see our answer to the similar request from Reviewer 1.

REVIEWERS' COMMENTS

Reviewer #1 (Remarks to the Author):

The manuscript has improved significantly, although it is more of a community resource than a primary research paper. I recommend publication.

I would suggest moving reviewer figures R1-3 to the actual manuscript for educational purposes and to show the extent of variability between cases. Based on the PCA analysis in Figure R1, I would have rejected the data, but it appears that other methods can still classify the samples (into patient vs. control), suggesting that meaningful differences can be extracted. I am sure I am not the only reader who could benefit from learning about these methods.

Reviewer #2 (Remarks to the Author):

Thank you for the revisions, all my concerns have been adequately addressed.

Reviewer #3 (Remarks to the Author):

The authors have made a very detailed effort to respond to the extensive reviewers comments. in some areas they have resolved issues but other areas remain a fundamental problem due to the low quality of RNA from the neurons in the FTD cases. while the authors claim the lack of numbers may preclude male/female differences this is now an important review criteria and some effort should have been made here even with low numbers.

Responses to reviewers' comments:

Reviewer #1 (Remarks to the Author):

The manuscript has improved significantly, although it is more of a community resource than a primary research paper. I recommend publication.

I would suggest moving reviewer figures R1-3 to the actual manuscript for educational purposes and to show the extent of variability between cases. Based on the PCA analysis in Figure R1, I would have rejected the data, but it appears that other methods can still classify the samples (into patient vs. control), suggesting that meaningful differences can be extracted. I am sure I am not the only reader who could benefit from learning about these methods.

We thank the Reviewer for his appreciation for the data set we generated. In response to the suggestion, we moved Fig. R1-3 to Supplementary Figs. 14-15.

Reviewer #2 (Remarks to the Author):

Thank you for the revisions, all my concerns have been adequately addressed.

Thank you for your appreciation of the value of our study, and for your helpful critiques.

Reviewer #3 (Remarks to the Author):

The authors have made a very detailed effort to respond to the extensive reviewers comments. in some areas they have resolved issues but other areas remain a fundamental problem due to the low quality of RNA from the neurons in the FTD cases. while the authors claim the lack of numbers may preclude male/female differences this is now an important review criteria and some effort should have been made here even with low numbers.

Thank you for this suggestion. As we previously reported, we attempted to analyze male and female donors separately and observed a substantial congruence within the DE genes identified in both sexes. Despite this, the small sample size for each sex in our study hindered us from achieving sufficient statistical power for this analysis. Consequently, we did not proceed with further exploration of sex-specific variations.